# CONTROLLABLE DIFFUSION-BASED GENERATION FOR MULTI-CHANNEL BIOLOGICAL DATA

**Haoran Zhang**
Department of Computer Science
University of Texas at Austin
Austin, TX 78712
hz6453@utexas.edu

**Mingyuan Zhou**
Department of Statistics
University of Texas at Austin
Austin, TX 78712
mingyuan.zhou@mccombs.utexas.edu

**Wesley Tansey**
Computational Oncology
Department of Epidemiology and Biostatistics
Memorial Sloan Kettering Cancer Center
New York, NY 10065
tanseyw@mskcc.org

## ABSTRACT

Biological profiling technologies, such as imaging mass cytometry (IMC) and spatial transcriptomics (ST), generate multi-channel data with strong spatial alignment and complex inter-channel relationships. Modeling such data requires generative frameworks that jointly model spatial structure and inter-channel dependencies and generalize across arbitrary subsets of observed and missing channels. Existing generative models typically assume low-dimensional inputs (e.g., RGB images) and rely on simple conditioning mechanisms that disrupt spatial correspondence and overlook inter-channel dependencies. This work proposes a unified multi-channel diffusion (MCD) framework for controllable generation of structured biological data with complex inter-channel relationships. Our model introduces two key innovations: (1) a hierarchical feature injection mechanism that enables multi-resolution conditioning on spatially aligned observed channels, and (2) two complementary channel attention modules to capture inter-channel relationships and recalibrate latent features. To support flexible conditioning and generalization to arbitrary sets of observed channels, we train the model using a random channel masking strategy, enabling it to reconstruct missing channels given any combination of observed channels as the spatial condition. We demonstrate state-of-the-art performance across both spatial and non-spatial biological data generation tasks, including imputation in spatial proteomics and clinical imaging, as well as gene-to-protein translation in single-cell datasets, and show strong generalizability to unseen conditional configurations.

## 1  INTRODUCTION

Recent advances in generative models enable structured generation, prediction, and imputation across various data domains (Rombach et al., 2022; Zhang et al., 2023). Specifically, diffusion models have demonstrated a remarkable capacity to generate high-fidelity samples in natural image and language generation tasks. In biology and medical domains, on the other hand, data acquisition is often constrained by experimental and clinical limitations. Spatial profiling technologies such as imaging mass cytometry (IMC) (Chang et al., 2017) and sequencing platforms like Xenium (Janesick et al., 2023) are expensive, time-consuming, and restricted by physical limitations that only allow measuring a limited number of signals of interest, e.g., around 50 proteins for IMC and 500 to 5000 genes for Xenium. Similarly, in clinical imaging, specific signals may be missing in practice due to patient motion, scan-time constraints, or acquisition artifacts (de Verdier et al., 2024). These constraints create a pressing need for biological data generation and imputation.

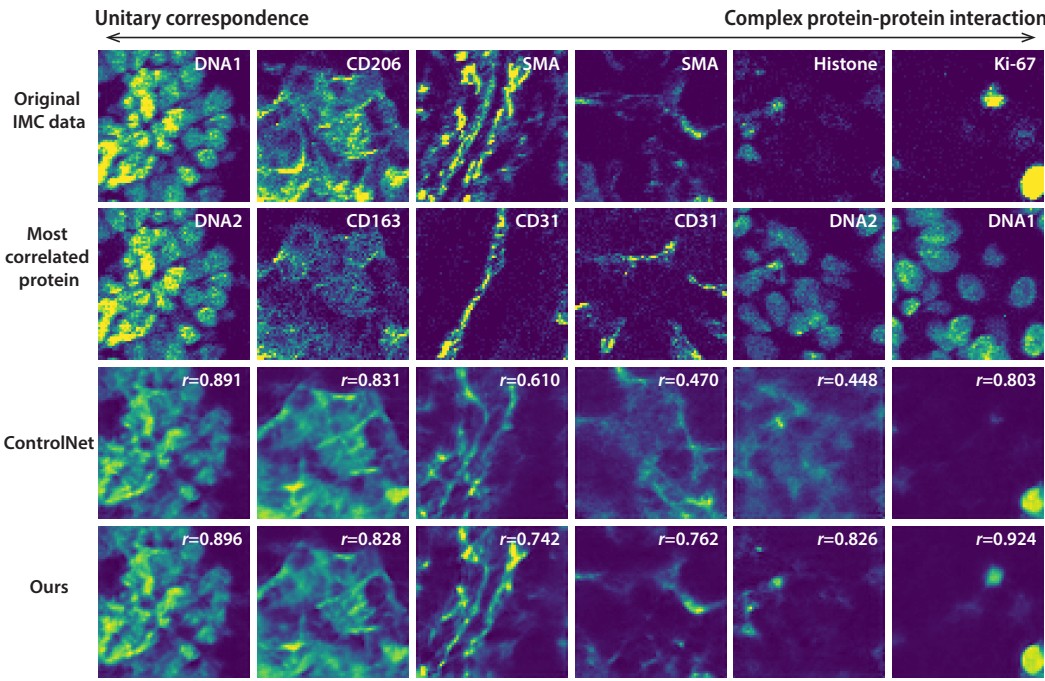

Figure 1: Visual comparison of samples generated by our model with samples generated by ControlNet-style counterpart. Without the channel attention modules and attention injection mechanisms, ControlNet-style generative models can handle simple channels with unitary inter-channel correlation, such as DNA1 (the right-most column), but easily fail in those with more complex protein-protein interactions, like tumor markers Ki-67 and $\alpha$SMA (columns 3-6).

A key characteristic of biological data is its multi-channel nature. For instance, in spatial profiling data, each channel is assigned to a specific molecule of interest (e.g., proteins $n \geq 30$ and genes $n \geq 100$), and each pixel (or cell) represents a spatially co-registered vector of biologically distinct signals. Unlike natural images with fixed and highly correlated RGB channels, biological channels often exhibit complex inter-channel dependencies. A generative framework that can handle high-dimensional biological data, maintain spatial alignment, accommodate random missing channels, and model diverse inter-channel relationships extends the application of existing technologies and recovers missing information in clinical acquisitions.

In biological data generation, the conditional inputs are typically observed signals of the same sample, which are spatially aligned with the target of generation. Such spatial alignment should be preserved throughout the generation process. Naive conditioning mechanisms, such as global embeddings and flattened concatenation, disrupt spatial correspondence, making them incompatible with the biological data generation objective. On the other hand, the number of channels is large in biological data and their interactions are highly context-dependent. Some channels co-localize only within specific spatial niches or cell types, while others may be mutually exclusive. Therefore, modeling such sparse, nonlinear, and asymmetric dependencies requires a generative framework that preserves spatial alignment and adapts to diverse, context-dependent channel semantics in biological data.

In this work, we introduce the multi-channel diffusion (MCD) framework for controllable multi-channel data generation. We demonstrate its performance and generalizability in tasks of generating biological profiling data with subsets of observable channels. Our method handles arbitrary combinations of observed and missing channels, maintains spatial alignment, and captures complex inter-channel dependencies. To achieve this, we propose three key components: (i) amortized conditioning with random channel masking, (ii) adaptive condition injection, and (iii) two complementary channel attention modules. We formalize an amortized conditioning loss and implement

it via random channel masking, enabling a single model to generalize across arbitrary conditional configurations of observed and missing channel combinations. Rather than pure element-wise addition, we reweight latent features in condition injection, allowing the model to dynamically gate spatial conditions. We also combine soft Squeeze-and-Excitation attention for per-sample global feature recalibration (used in condition injection) together with a transformer-style channel attention module inside UNet blocks to model cross-channel dependencies. Ablation studies confirm that each component improves data generation quality. Together, these components enable our model to generate high-fidelity multi-channel biological data across various condition-target combinations within a unified diffusion framework. Our model achieves state-of-the-art results on both spatial and non-spatial biological data generation tasks and supports test-time controllability for arbitrary conditional configurations.

## 2 BACKGROUND

### 2.1 DIFFUSION MODELS

Diffusion models are a class of generative models that approximate the data distribution by modeling the gradient of its log-density (Song & Ermon, 2019). This is achieved by coupling a forward process that progressively perturbs the data with a reverse process that learns to recover the original. The forward process transforms data $\mathbf{x}_0$ into a noise distribution $\mathbf{x}_T$ over $T$ timesteps by injecting noise from a known distribution like a Gaussian (Sohl-Dickstein et al., 2015), $q(\mathbf{x}_t \mid \mathbf{x}_0) = \mathcal{N}(\mathbf{x}_t \mid \sqrt{\alpha_t}\mathbf{x}_0, (1 - \alpha_t)\mathbf{I})$, where $\alpha_t$ is a noise schedule parameterizing the variance at each timestep $t$. As $t$ approaches $T$, the data distribution converges to a simple prior, such as a standard Gaussian. The reverse process reconstructs data from noise by learning to reverse the corruption introduced during the forward process. This is achieved through a neural network $\epsilon_\theta(x_t, t)$, trained to predict the noise $\epsilon$ added at each timestep (Ho et al., 2020). The denoising process is guided by the objective $\mathbb{E}_{\mathbf{x}_0, \epsilon, t} \left[ \|\epsilon - \epsilon_\theta(\mathbf{x}_t, t)\|^2 \right]$, where $\epsilon \sim \mathcal{N}(0, \mathbf{I})$ is the Gaussian noise injected by the forward process. By minimizing this objective, the model learns to reconstruct $x_0$ from the noisy $x_t$ at any timestep $t$. This denoising approach is deeply connected to score matching. Specifically, the noise prediction $\epsilon_\theta(x_t, t)$ can be used to compute the gradient of the log-density (i.e., score function) as $\nabla_{\mathbf{x}_t} \log q(\mathbf{x}_t) \propto -\frac{\epsilon_\theta(\mathbf{x}_t, t)}{\sqrt{1 - \alpha_t}}$. Thus, accurately predicting the noise is equivalent to estimating the score function $\nabla_{\mathbf{x}_t} \log q(\mathbf{x}_t)$, which guides the reverse process. This insight bridges denoising and score matching, making the reverse process a refinement procedure that progressively moves noisy samples back to the data manifold (Song & Ermon, 2019).

### 2.2 STRUCTURED MULTI-CHANNEL IMPUTATION

Image imputation is a classical problem in computer vision and generative modeling, where the objective is to recover missing or corrupted parts of an image given the observed context. Formally, given an observation $c$, classical imputation methods aim to estimate the full image $x$ by modeling the conditional distribution $p(\mathbf{x} \mid c)$. This problem has been extensively studied in the context of natural images, where $\mathbf{x} \in \mathbb{R}^{3 \times H \times W}$, and inpainting primarily relies on spatial continuity within the image domain (Chan & Shen, 2001).

We generalize this problem to the setting of structured multi-channel data, where each channel corresponds to a semantically distinct signal—such as a spectral band, a protein marker, or a gene expression. Let $c \in \mathbb{R}^{C_o \times H \times W}$ denote the observed data, and let $\mathbf{x} \in \mathbb{R}^{C \times H \times W}$ denote the full data, where $C = C_o + C_m \geq 1$, $C_o \geq 1$, and $C_m \geq 0$ denote the number of total, observed, and missing channels respectively. Note that when $C_m = 0$ and $C_o = C = 3$, this formulation reduces to the classical RGB image inpainting problem ($C = 1$ for the grayscale case). When $C_m = 3$ (RGB) and $C_o = 1$ (grayscale), it reduces to the classical image colorization problem. This general formulation applies across a broad range of problems at different resolutions. When $H = W = 1$, the input reduces to a high-dimensional vector, making this formulation applicable to non-spatial data imputation, such as single-cell data, where one can predict single-cell protein expression from single-cell RNA sequencing (scRNA-seq) data. When $H, W > 1$, the formulation supports spatially structured imaging tasks, such as IMC channel prediction, where both local morphology and global tissue organization contribute to the reconstruction target.

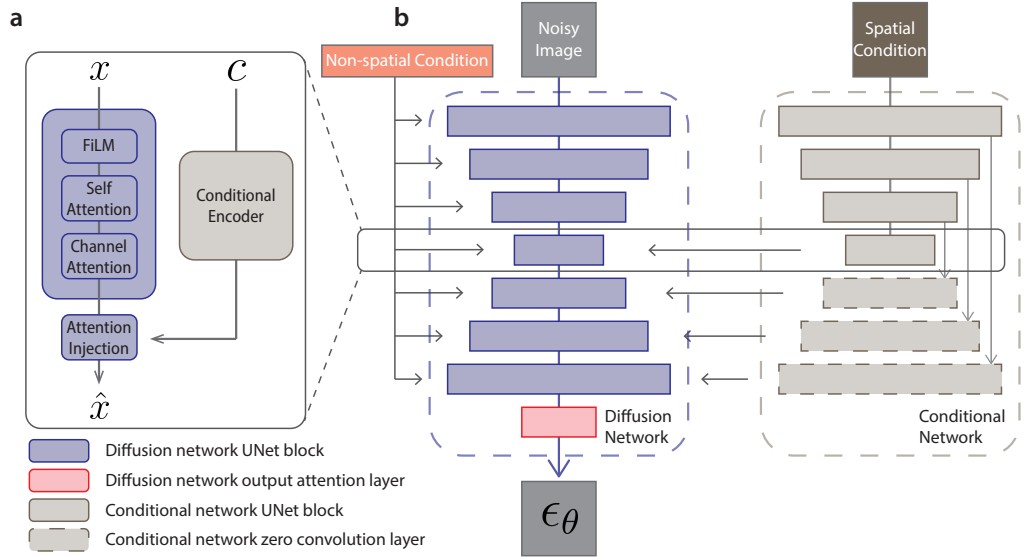

Figure 2: Overview of the proposed MCD for multi-channel data generation. (a) Custom channel attention: UNet block with channel attention module and attention injection block that model the inter-channel relationships. (b) Hierarchical feature injection: a parallel conditional network encodes the spatial condition, producing features at different resolutions that are injected into the corresponding UNet block in the diffusion network.

## 3 METHODS

We propose a diffusion-based generative framework, MCD, for multi-channel biological data that conditions on arbitrary subsets of observable channels. Let $\mathbf{x} \in \mathbb{R}^{C \times H \times W}$ denote the full-channel data and $c \in \mathbb{R}^{C_o \times H \times W}$ an observed subset, where $\mathbf{x}$ and $c$ are spatially aligned. The goal is to learn the conditional distribution $p(\mathbf{x}|c)$ in a way that respects the spatial structure of the data while remaining flexible to any condition $c$. This requires resolving several intertwined challenges: 1) the generated targets should be spatially aligned with the conditions; 2) the conditions should be modeled as structured, multi-resolution information; 3) correlations among biological channels are complex; and 4) the model must generalize across arbitrary condition–target combinations at test time. To address these challenges, MCD employs a dual-network architecture consisting of a diffusion network that denoises the noisy target $\mathbf{x}_t$ and a conditional network that encodes the observed channels $c$ (Figure 2b). At each resolution level $\ell$, the diffusion encoder produces features $D_\ell(x_t)$ while the contextual encoder provides aligned features $E_\ell(c)$. These conditional representations are hierarchically injected into the diffusion network at corresponding resolution $\ell$ to ensure alignment and effective spatial conditioning. Channel attention modules capture inter-channel dependencies, and random channel masking training ensures generalizability across channels.

### 3.1 SPATIALLY ALIGNED FEATURE INJECTION FOR STRUCTURED CONDITIONING

Generating realistic multi-channel biological data requires preserving spatial correspondence with the observed channels and incorporating the conditioning signal that guides both local details and global structure. To achieve this, we introduce a hierarchical feature injection mechanism. At each resolution level $\ell$, contextual features $E_\ell(c)$ are spatially aligned with and injected into the diffusion features $D_\ell(x_t)$ through

$$\mathbf{z}_\ell = D_\ell(\mathbf{x}_t) + \text{SE}(E_\ell(c)) \tag{1}$$

allowing for the flexible injection of spatial conditions ("Attention Injection" block in Figure 2a). The $\text{SE}(\cdot)$ is the Squeeze-and-Excitation block that serves as a soft channel attention mechanism (see section 3.3 for detail) and selectively injects the condition while the spatial alignment is preserved. $\text{SE}(E_\ell(c))$ maintains spatial dimensions and applies channel-wise gating to the condition

feature map before element-wise addition. This design allows the model to condition not just on a fixed global representation of $c$ but on a series of contextual features $\{E_\ell(c)\}_{\ell=1}^L$ that vary across resolution. This setup reflects the intuition that certain patterns in $\mathbf{x}$, e.g., global motifs versus local structures, may depend on different aspects of $c$. Early encoder layers focus more on the local structures, while late layers draw high-level, global structures.

## 3.2 TRAINING WITH RANDOM CHANNEL MASKING

Different experiments may measure different sets of signals. Rather than training separate models to predict individual channels, our goal is to produce full multi-channel outputs (i.e., the complete protein panel). Constructing our model to condition on any subset enables us to train it across multiple datasets and flexibly apply it to each. To create a unified model, we propose a random channel masking strategy during training (Algorithm 1). In each iteration, we randomly sample a subset of channels $S_o \subset \{1, \ldots, C\}$ as observed, and mask out the remaining $S_m = \{1, \ldots, C\} \backslash S_o$. The masked subset $\mathbf{x}_{S_m}$ is zeroed out in spatial condition $c$, and the model reconstructs the full panel $\mathbf{x} \in \mathbb{R}^{C \times H \times W}$. We optimize the standard EDM objective (Karras et al., 2022) on the full-channel target $\mathbf{x}$, and the binary mask is only applied to the conditions.

---

**Algorithm 1** Random Channel Masking

---

**Require:** Full data $\mathbf{x} \in \mathbb{R}^{C \times H \times W}$, masking probability $p$
1: **for** each training iteration **do**
2:     Sample observed set $S_o \subset \{1, \ldots, C\}$, where $I_{i \in S_o} \sim Bern(p)$
3:     Construct condition $c$ such that

$$c_i = \begin{cases} \mathbf{x}_i, & \text{if } i \in S_o \\ 0, & \text{otherwise} \end{cases}$$

4:     Diffusion step with target $\mathbf{x}$ and condition $c$
5: **end for**

---

Randomly varying $S_o$ during training encourages the model to learn conditional generation under diverse partial contexts. As a result, the trained model can generalize to arbitrary combinations of observed and missing channels at test time, including unseen configurations. At the same time, because the model always reconstructs all channels, it avoids the need for channel-specific heads or separate training for each channel, enabling comprehensive downstream analysis and one-stop training.

This training procedure implicitly defines an amortized conditional learning objective:

$$\mathbb{E}_{c \sim P(c)} \mathbb{E}_{t, \mathbf{x}_0, \epsilon} \left[ ||\epsilon - \epsilon_\theta(\mathbf{x}_t, t, c)||^2 \right] \tag{2}$$

where $p(c)$, with $c \in \mathcal{C}$, denotes the distribution of conditional configuration (i.e., combinations of observable channels) over the conditional space $\mathcal{C}$. Minimizing eq. (2) encourages the model to learn a single amortized estimator $\epsilon_\theta(\mathbf{x}_t, t, c)$ such that $\epsilon_\theta(\mathbf{x}_t, t, c) \approx \nabla_{\mathbf{x}_t} \log p(\mathbf{x}_t|c)$ for arbitrary condition $c$. Under standard diffusion assumptions, random channel masking ensures that the model is trained on random conditional subsets, and the learned model generalizes across configurations and implicitly models the family of conditional distributions $\{p(x|c), c \in \mathcal{C}\}$. Therefore, this formulation frames random masking as an amortized inference over the condition space. Although not formally addressed, prior work has demonstrated its effectiveness across various domains (Gershman & Goodman, 2014; Marino et al., 2018). We provide theoretical justification in Appendix A. Empirically, we observe that this strategy enables robust generalization to unseen conditional configurations at test time.

## 3.3 CHANNEL ATTENTION FOR STRUCTURED FEATURE MODULATION

With random channel masking, our model aims to handle arbitrary conditional configurations, where the observed channels may vary across samples and tasks. Since all conditions $c$ are encoded with the same conditional network, the model needs to select and reweight latent features adaptively

depending on the conditions. To enable dynamic feature recalibration in the latent space, we introduce two complementary channel attention modules with distinct designs, each focusing on specific modeling goals.

The first is a lightweight attention mechanism ("Attention Injection" block in Figure 2a) inspired by the Squeeze-and-Excitation (SE) network (Hu et al., 2018). Given a latent feature map $\mathbf{z} \in \mathbb{R}^{D \times H \times W}$, we compute

$$
\begin{aligned}
\boldsymbol{\alpha} &= \mathrm{GAP}(\mathbf{z}) \\
\mathbf{w} &= \sigma \left( W_2 \cdot \phi(W_1 \cdot \boldsymbol{\alpha}) \right) \\
\mathbf{z}' &= \mathbf{w} \cdot \mathbf{z}
\end{aligned}
\tag{3}
$$

where $\phi$ is a non-linear activation function (e.g., ReLU), GAP stands for global average pooling, and $\sigma$ is the sigmoid function. This approach scales each latent channel by a learned weight conditioned on the global context and enables a per-channel feature reweighting that aligns with the dynamic feature selection requirement for multi-scale conditional injection (Section 3.1). This SE-based attention mechanism is lightweight and stabilizes hierarchical feature injection.

On the other hand, we also perform a self-attention over all latent channels ("Channel Attention" blocks in Figure 2a)

$$
\begin{aligned}
\mathbf{x}_{\text{flat}} &\in \mathbb{R}^{D \times N}, \quad N = H \times W, \\
Q &= \mathbf{x}_{\text{flat}} W_Q, \quad K = \mathbf{x}_{\text{flat}} W_K, \quad V = \mathbf{x}_{\text{flat}} W_V
\end{aligned}
\tag{4}
$$

$$
A = \mathrm{softmax}\left( \frac{QK^\top}{\sqrt{d}} \right), \quad \mathbf{x}'_{\text{flat}} = AV
\tag{5}
$$

Compared to SE channel attention, this design is more expressive and can capture higher-order dependencies among latent channels. This feature makes the transformer-based channel attention particularly useful in UNet blocks, where the model infers missing information across latent channels. Rather than reweighting channels independently, it allows information to propagate across channels through learned interactions. This is particularly important in biological data, where channel relationships are often nonlinear, asymmetric, and context-dependent.

MCD includes an additional soft channel attention block, in the same way as Equation (3), at the final stage of the model, which maps the latent channels to data channels. Let the final latent representation before projection be $\mathbf{z} \in \mathbb{R}^{D \times H \times W}$, where $D$ is the number of latent channels. The standard diffusion model produces $\mathbf{y} \in \mathbb{R}^{C \times H \times W}$ with a single output layer on $\mathbf{z}$. To model cross-channel dependencies, we add an additional layer that computes

$$
\hat{\mathbf{y}}_{\text{attn}} = \mathbf{y} + \mathrm{Conv1}(\mathrm{SE}(\mathbf{y}))
\tag{6}
$$

While latent-space attention captures dependencies among hidden features, the output-space attention provides a final channel-wise recalibration in the data space.

Together, these mechanisms provide both adaptive gating (SE) and structured channel interaction modeling (self-attention), enabling robust feature modulation across heterogeneous conditional configurations and improving performance on tasks involving complex, structured channel relationships.

## 4  RELATED WORK

**Image inpainting with conditional diffusion models.** Recent work has applied diffusion-based generative models to image inpainting and completion tasks (Song & Ermon, 2019; Saharia et al., 2022). These models primarily work on natural and grayscale images, where the number of channels is limited ($n \leq 3$), and the goal is to reconstruct spatially masked regions based on the surrounding context. These models often take conditionals like class labels (Dhariwal & Nichol, 2021), text embeddings (Ramesh et al., 2022; Nichol et al., 2022), or segmentation maps (Rombach et al., 2022). These conditionals are typically injected via concatenation at the input or embedding injection via FiLM modulation. This approach ignores the potential spatial alignment between the conditionals and the generative target. More recent approaches like ControlNet (Zhang et al., 2023) and BrushNet (Ju et al., 2024) introduce multiscale conditioning mechanisms that are spatially aligned and applied post hoc to pre-trained Stable Diffusion models. While these methods preserve spatial alignment,

they assume low-dimensional inputs and do not address the challenges posed by high-dimensional structured data with intricate inter-channel dependencies. Moreover, because their conditioning modules are trained separately from the core generative model, they lack end-to-end coordination between condition encoding and generation.

**Dynamic guidance for conditional diffusion.** Classifier-free guidance (CFG) (Ho & Salimans, 2021) introduces a flexible training scheme for conditional diffusion models by randomly dropping conditioning inputs and jointly training the model on both conditional and unconditional objectives. While originally developed for low-dimensional conditioning signals such as class labels, the core idea can be generalized: using input masking during training to enable flexible guidance at test time. We adopt this principle in the form of random-masking guidance, where the spatial conditions are randomly masked during training. Unlike CFG, which interpolates between conditional and unconditional predictions at inference time, our method samples conditioning subsets during training to learn a single unified conditional model. Specifically, the model observes a random subset of input channels and is trained to reconstruct the full panel. This dynamic masking encourages the model to generalize across different conditionals, making it suitable for tasks where the available condition varies across samples. This approach enables multi-channel prediction and supports generation under arbitrary conditioning subsets without retraining or architectural changes.

**Channel attention mechanisms.** While channel-wise attention has been explored in vision architectures like SENet (Hu et al., 2018), most diffusion-based models focus on spatial attention and overlook channel-wise relationships. However, this becomes a significant limitation in multi-channel biological data. A few recent works in diffusion-based colorization have begun to explore this direction: FCNet (Zhu et al., 2024) introduces spatially decoupled color representations tailored to facial regions, and ColorPeel (Butt et al., 2024) learns disentangled geometric shapes and color embeddings in latent space. These models can be thought of as coarse region-level attention over RGB channels and are not designed for tasks involving dozens or hundreds of semantically distinct channels.

## 5 EXPERIMENTS

### 5.1 SINGLE-CELL MODALITY PREDICTION

To evaluate MCD's performance in non-spatial biological data, we begin with the standard CITE-seq benchmark of predicting protein expression from paired scRNA-seq, where the targets are single-cell protein expression, and the conditions are scRNA-seq expression. The model is trained without random channel masking since both the target and the condition channels are fixed. We compare MCD to a suite of state-of-the-art methods: Kernel Ridge Regression (Guanlab-dengkw) (Lance et al., 2022), MultiVI (Ashuach et al., 2023), GLUE (Cao & Gao, 2022), scMM (Minoura et al., 2021), and UnitedNet (Tang et al., 2024). Across four datasets: peripheral blood mononuclear cells (PBMC, Hao et al. (2021)), cord blood mononuclear cells (CBMC, Stoeckius et al. (2017)), bone marrow mononuclear cells (BMMC, Lance et al. (2022)), and hematopoietic stem and progenitor cells (HSPC, Nestorowa et al. (2016)), our model consistently outperforms the existing single-cell modality translation methods (Table 1). Performance is reported as average Pearson correlation between predicted and measured expression levels across proteins ($r_p$) and cells ($r_c$). Importantly, our framework achieves the highest protein-level correlation ($r_p$), the more biologically relevant metric, in every dataset. This reflects the model's ability to capture gene–protein relationships at scale.

We also distilled our diffusion model with SiD (Zhou et al., 2023) into a one-step generation variant, which retains comparable accuracy while reducing inference cost by two orders of magnitude. This suggests that MCD not only yields high-accuracy generation but is also ready for large-scale deployment.

Table 1: Benchmarking our multimodal diffusion approach against existing modality prediction methods on the gene-to-protein prediction task on four (PBMC, CBMC, BMMC, and HSPCs) datasets. $r_c$ shows the cell-wise correlation with the ground truth, and $r_p$ shows the protein-level correlation.

| Method | PBMC | | CBMC | | BMMC | | HSPC | |
|---|---|---|---|---|---|---|---|---|
| | $r_c$ | $r_p$ | $r_c$ | $r_p$ | $r_c$ | $r_p$ | $r_c$ | $r_p$ |
| KRR (Guanlab-dengkw) | **0.908** | 0.646 | 0.863 | 0.006 | 0.870 | 0.094 | 0.820 | 0.059 |
| MultiVI | 0.088 | 0.069 | 0.103 | 0.032 | 0.082 | 0.054 | 0.045 | 0.035 |
| GLUE | 0.659 | -0.004 | 0.623 | 0.054 | 0.589 | 0.012 | 0.549 | 0.024 |
| UnitedNet | 0.870 | 0.518 | 0.377 | 0.628 | 0.625 | 0.634 | 0.310 | 0.436 |
| scMM | 0.793 | 0.521 | 0.736 | 0.517 | 0.853 | 0.625 | 0.724 | 0.598 |
| Ours (500 steps) | 0.880 | **0.673** | **0.962** | **0.763** | **0.879** | **0.685** | **0.865** | **0.647** |
| Ours + SiD (1 step) | 0.874 | 0.672 | **0.962** | 0.759 | 0.875 | 0.682 | **0.865** | 0.642 |

## 5.2 SPATIAL DATA IMPUTATION

### 5.2.1 SINGLE DATASET IMPUTATION

To evaluate the model's performance in spatial biological data, we apply it to spatial proteomics data from imaging mass cytometry (IMC), where each pixel corresponds to spatially co-registered multi-protein expression signals. We evaluate our method on two IMC datasets: a lung cancer cohort (Yoffe et al., 2025) (8 patients) and a breast cancer cohort (Jackson et al., 2020) (6 patients), with 50 and 35 co-registered protein channels, respectively. The spatial data are normalized per channel and partitioned into $64 \times 64$ patches, which are further downsampled to $16 \times 16$ by a pretrained encoder. For each dataset, we compare our generation to two data-specific baselines: the protein with the highest spatial correlation and kernel ridge regression, as well as structured conditional diffusion models, ControlNet (Zhang et al., 2023). We also include domain-specific methods, STEM (Zhu et al., 2025) and Virtues (Wenckstern et al., 2025). We evaluate under two protocols: (1) single-channel, where a dedicated model is trained per target channel, and (2) multi-channel, where a single model is trained with random channel masking. In the single-channel prediction setup, we mask one protein channel at a time as the target and use the remaining channels as the spatial condition. On the other hand, in the multi-channel prediction setup, the observed channels are randomly selected by independent $Bern(0.9)$ random variables, and the remaining channels are set to zero. The model is trained to reconstruct the full-channel data. To explicitly evaluate generalization across unseen conditional configurations, we ensure that the masking used at test time includes channel subsets not observed during training. Successful reconstruction under previously unseen masking patterns provides evidence for true amortized generalization across the conditional space rather than memorization of fixed condition–target configurations.

Table 2 shows our method outperforms all existing approaches on both IMC datasets. Most baseline models fail to outperform the best linear predictor, as well as the most correlated individual observed protein. By contrast, our method consistently outperforms the baselines across each protein channel. Fixing the missing channel during training improves performance in the single-channel prediction mode compared to the multi-channel setup, which reflects the capacity trade-off in amortized multi-task learning. When the target channel is fixed during training, the model can dedicate its capacity to modeling a single conditional distribution. In contrast, the multi-channel model amortizes its capacity across all possible missing-channel configurations. Despite this broader objective, the multi-channel model still outperforms all baselines, demonstrating that the random masking strategy enables strong generalization with high predictive accuracy. These results validate the effectiveness of random-masking guidance and support practical use cases where multi-channel prediction is required. Visualization of the generated IMC samples (Figure 6), ablation studies (Appendix B.2), and quantifications of uncertainty and correlations between generated channels (Appendix B.3) are provided in the appendix.

Table 2: Comparison of predictive correlation across different methods in spatial prediction tasks. Our method outperforms the diffusion, single-protein, and linear baselines, and diffusion based ControlNet, domain-specific models Stem and Virtues.

| Method | Breast | Lung |
|---|---|---|
| Most correlated protein | 0.481 | 0.506 |
| Kernel ridge regression | 0.489 | 0.527 |
| Virtues (Wenckstern et al., 2025) | 0.398 | 0.425 |
| Stem (Zhu et al., 2025) | 0.403 | 0.475 |
| ControlNet (Zhang et al., 2023) | 0.452 | 0.537 |
| Ours (single channel) | **0.667** | **0.703** |
| Ours (multi channel) | 0.596 | 0.647 |

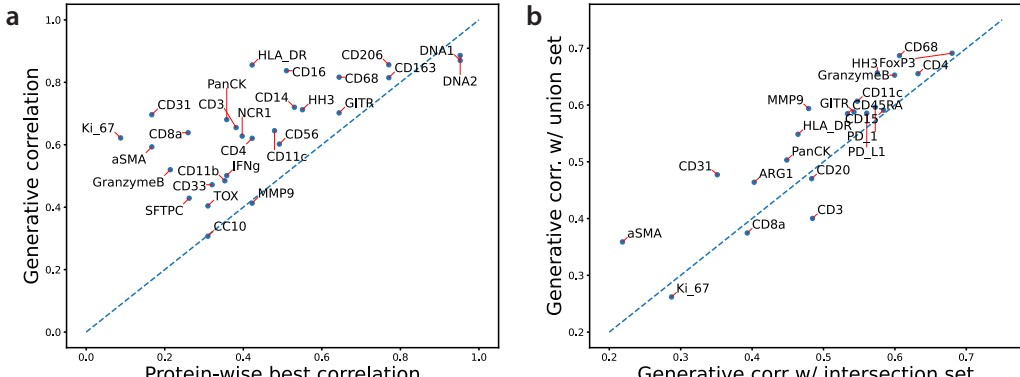

Figure 3: (a) Benchmarking generative performance (Pearson's $r$) against the most similar protein on the lung cancer dataset. (b) Cross-dataset generalization study results. Using the union of all protein channels, even those missing from the test dataset, consistently outperforms using only the overlapping set of protein channels.

### 5.2.2 MULTI-DATASET GENERALIZATION

We further test our model's ability to learn from multiple datasets with partially overlapped protein channels. Specifically, we combine the lung and breast cancer IMC datasets. The two datasets share 23 proteins in common. We evaluate our model under two training schemes: (1) an intersection setting, where we keep only the proteins observed in both datasets, and (2) a union setting, where we include the union of all protein channels and zero-pad missing channels in each dataset. Both setups share the same model architecture and training strategy with random masking, and predict the intersection set at test time.

As shown in Figure 3b, the union setup consistently outperforms the intersection counterpart, achieving a higher average Pearson correlation in sample generation. The union setup has broader coverage and enables the model to learn a richer set of inter-channel dependencies and improves its robustness to missing data. This result empirically validates that random channel masking enables principled joint learning under heterogeneous supervision. By zero-padding unobserved channels and sampling conditional subsets during training, the model does not require perfect channel alignment across datasets. Instead, it learns to infer conditional structure from partially overlapping panels. The performance in the union setting demonstrates MCD and random channel masking as a principled method for multi-dataset integration with empirical advantages.

## 5.3 MRI MODALITY SYNTHESIS

To test the generalizability of our framework, we evaluate on the BraTS benchmark (de Verdier et al., 2024) for missing–modality magnetic resonance imaging (MRI) synthesis, where the goal is to reconstruct unacquired MR sequences given observed ones. This task is clinically relevant in brain tumor imaging, as missing scans are common in practice due to acquisition time constraints and patient-specific factors. We follow the BraTS guidelines, using tissue structural similarity (SSIM) for image fidelity and Dice coefficient (DICE) for downstream tumor segmentation quality as evaluation metrics.

Table 3 summarizes results across SOTA baselines, including Pix2pix (Isola et al., 2017), HF-GAN (Cho et al., 2024), and SwinUNETR (Pang et al., 2025), where the latter two are the winner and the runner-up of the BraTS 2024 challenge, respectively. Our method achieves the highest scores across both SSIM and DICE, improving both the structural similarity and clinical utility. Gains in DICE demonstrate that our approach not only reconstructs visually faithful images but also preserves tumor structures critical for segmentation. Compared with HF-GAN and SwinUNETR, which represent state-of-the-art GAN- and transformer-based baselines, our model shows consistent improvements in SSIM and DICE. Visualization of the generated MRI samples is provided in the appendix (Figure 7).

Table 3: Comparison of segmentation accuracy and structural similarity across different synthesis methods with DICE and SSIM scores across tumor, tissue, and global regions.

| Method | DICE $\uparrow$ | $SSIM_{tumor} \uparrow$ | $SSIM_{healthy} \uparrow$ | $SSIM_{tissue} \uparrow$ | $SSIM_{global} \uparrow$ |
|---|---|---|---|---|---|
| pix2pix | 0.549 | 0.719 | 0.570 | 0.583 | 0.807 |
| HF-GAN | 0.714 | 0.761 | 0.604 | 0.615 | 0.919 |
| SwinUNETR | 0.709 | 0.759 | 0.628 | 0.637 | 0.916 |
| MCD | **0.738** | **0.774** | **0.631** | **0.643** | **0.928** |

## 6 CONCLUSION

We introduce MCD as a diffusion-based generative model for the controllable generation of multi-channel biological data, which is generalizable to arbitrary combinations of observed and missing channels. By combining amortized conditioning via random channel masking, hierarchically aligned feature injection, and structured channel attention, MCD captures complex inter-channel dependencies while maintaining spatial alignment. Our channel masking and simulation-based analyses highlight the effectiveness of MCD, and our experiments on MRI modality synthesis demonstrate that the method can provide tangible benefits in real-world applications, such as recovering missing clinical signals to support downstream analyses.

MCD offers an *in silico* extension to the experimentally constrained profiling panel, and recovers corrupted or incomplete acquisitions. Together, these properties suggest that this framework can serve as a scalable backbone for structured biological data modeling. In this sense, MCD provides a step toward a foundation model for spatial and multi-modal biological profiling, where a single generative prior supports imputation, synthesis, and multi-dataset integration. While this work focuses on methodological development and demonstrates the usefulness of the proposed framework in biological image generation tasks, future directions include scaling to larger and more diverse spatial cohorts, incorporating richer biological priors, and facilitating biologically meaningful discovery.

REPRODUCIBILITY STATEMENT

The complete theoretical justification and assumption needed are provided in Appendix A. Code is available at https://github.com/tansey-lab/MCD.

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

# A    THEORETICAL JUSTIFICATION OF RANDOM MASKING

We provide a theoretical justification for our random masking training strategy. In appendix A.1, we show that the batch-wise empirical amortized loss upper bounds the true amortized loss. In appendix A.2, we demonstrate that the model's performance in an arbitrary condition configuration is controlled, with high probability, under the empirical assumption of variance. We define the amortized loss

$$
\begin{aligned}
L &= \mathbb{E}_c \mathbb{E}_{t,x_0,\epsilon} \left[ \ell(\epsilon, \epsilon_\theta(x_t, t, c)) \right] \\
&= \mathbb{E}_c \mathbb{E}_{t,x_0,\epsilon} \left[ ||\epsilon - \epsilon_\theta(x_t, t, c)||^2 \right]
\end{aligned}
\tag{7}
$$

and the empirical loss

$$
\hat{L}_N = \frac{1}{N} \sum_{i=1}^{N} \mathbb{E}_{t,x_0,\epsilon} \left[ ||\epsilon - \epsilon_\theta(x_t, t, c)||^2 \right]
\tag{8}
$$

under the bounded loss assumption.

**Assumption 1.** *(**Bounded loss**) The loss $\mathbb{E}_{t,x_0,\epsilon}\ell(\epsilon, \epsilon_\theta(x_t, t, c_i)) : \mathcal{D} \times \mathcal{D} \mapsto [0, d]$ asymptotically for trained $\epsilon_\theta$ by minimizing $\hat{L}_N$.*

Since $\epsilon \sim \mathcal{N}(\mathbf{0}, \mathbf{I})$ has an expected squared norm proportional to its dimension, the per-sample squared loss is bounded in expectation. Specifically, for normalized input and standard Gaussian noise, the loss is upper bounded by the variance of $\epsilon$, i.e., $\mathbb{E}[||\epsilon||^2] = D := CHW$, and significantly lower as the model converges. Therefore, for the following justifications, we assume the loss lies within a fixed scale $d \leq D$ asymptotically, and we will provide empirical evidence in appendix B.1

## A.1    EMPIRICAL BOUND OF THE AMORTIZED LOSS

**Theorem 1.** *[**Bartlett and Mendelson '02**] Consider a loss function $\mathcal{L} : Y \times A \mapsto [0, 1]$ and a dominating cost function $\phi : Y \times A \mapsto [0, 1]$. Let $F$ be a class of functions mapping from $X$ to $A$ and let $(X_i, Y_i)_{i=1}^{n}$ be independently selected according to the probability measure $P$. Then, for any integer $n$ and any $0 < \delta < 1$, with probability at least $1 - \delta$ over samples of length $n$, every $f$ in $F$ satisfies*

$$
\mathbb{E} \left[ \mathcal{L}(Y, f(X)) \right] \leq \hat{\mathbb{E}}_n \left[ \phi(Y, f(X)) \right] + R_n(\tilde{\phi} \circ F) + \sqrt{\frac{8 \ln(2/\delta)}{n}}
\tag{9}
$$

*where $\tilde{\phi} \circ F = \{(x, y) \mapsto \phi(y, f(x)) - \phi(y, 0) : f \in F\}$, and $R_n$ is the Rademacher complexity.*

Theorem 1 follows McDiarmid's inequality, and the detailed proof is provided by (Bartlett & Mendelson, 2002).

**Corollary 1.** *Under the bounded-loss assumption, with probability at least $1-\delta$, $L$ is upper-bounded by $\hat{L}_N$ up to an additive constant.*

*Proof.* Define function class $F_\theta := \{\epsilon_\theta(x_t, t, c)\}$ for the trainable diffusion model. Let $\phi = \mathcal{L} = \mathbb{E}_{t,x_0,\epsilon}\ell_{\text{norm}}(\epsilon, f_c) : \mathcal{D} \times \mathcal{D} \mapsto [0, 1]$, where $\ell_{\text{norm}} = \frac{1}{d}\ell$ is the normalized squared loss and $\mathcal{D}$ is the data space such that $x, \epsilon \in \mathcal{D}$, and eq. (9) becomes

$$
d\mathbb{E}_c \mathbb{E}_{t,x_0,\epsilon} \left[ \ell_{\text{norm}}(\epsilon, f_c(x_t, t)) \right] \leq \frac{d}{N} \sum_{i=1}^{N} \mathbb{E}_{t,x_0,\epsilon} \left[ \ell_{\text{norm}}(\epsilon, f_{c_i}(x_t, t)) \right] + dR_N(\ell_{\text{norm}} \circ F_\theta) + d\sqrt{\frac{8 \ln(2/\delta)}{N}}
$$

$$
\mathbb{E}_c \mathbb{E}_{t,x_0,\epsilon} \left[ d\ell_{\text{norm}}(\epsilon, f_c(x_t, t)) \right] \leq \frac{1}{N} \sum_{i=1}^{N} \mathbb{E}_{t,x_0,\epsilon} \left[ d\ell_{\text{norm}}(\epsilon, f_{c_i}(x_t, t)) \right] + R_N(d\ell_{\text{norm}} \circ F_\theta) + d\sqrt{\frac{8 \ln(2/\delta)}{N}}
$$

$$
\mathbb{E}_c \mathbb{E}_{t,x_0,\epsilon} \left[ \ell(\epsilon, f_c(x_t, t)) \right] \leq \frac{1}{N} \sum_{i=1}^{N} \mathbb{E}_{t,x_0,\epsilon} \left[ \ell(\epsilon, f_{c_i}(x_t, t)) \right] + R_N(\ell \circ F_\theta) + d\sqrt{\frac{8 \ln(2/\delta)}{N}}
$$

$$
L(\epsilon_\theta) \leq \hat{L}_N(\epsilon_\theta) + C(\delta)
$$

$$
\tag{10}
$$

where $C(\delta) = R_N(\ell \circ F) + d\sqrt{\frac{8\ln(2/\delta)}{n}}$. By Bartlett and Mendelson Bartlett & Mendelson (2002), $R_N(dF_\theta) = dR_n(F_\theta)$ and $R_N(\ell \circ F_\theta) \leq L_\ell R_n(F_\theta)$ where $\ell$ is Lipschitz continuous with constant $L_\ell$. Under the bounded-loss assumption, reasonable loss functions, including the squared loss we use, satisfy the Lipschitz condition on $[0, d]$, and $C(\delta) \leq L_\ell R_N(F_\theta) + \sqrt{\frac{8\ln(2/\delta)}{n}}$, where $R_N(F_\theta)$ is determined by the model architecture and $C(\delta)$ is independent of $f$ once the function class $F_\theta$ is fixed. Empirically, we can find $d < 1$ as shown in fig. 4. Therefore, the true amortized loss is upper-bounded by the empirical loss up to an additive constant. $\square$

Similarly, one can get the lower bound of the same fashion

$$\hat{L}_N(\epsilon_\theta) \leq L(\epsilon_\theta) + C(\delta) \tag{11}$$

and similar results can also be drawn from Corollary 18 from the work by Cortes et al. (2019). Therefore, we show that minimizing our empirical loss also minimizes the true amortized loss.

## A.2 Condition-specific bound by amortized loss

**Lemma 1.** *Assuming relatively low variance, i.e. $\sigma \leq \mu$, with high probability, the per-condition loss is of the same magnitude as the amortized loss $\mu$, i.e. $\mathbb{E}_{t,x_0,\epsilon}[\ell(\epsilon, f_c(x_t, t))] = \mathcal{O}(\mu)$*

*Proof.* Define the condition-specific loss

$$L_c = \mathbb{E}_{t,x_0,\epsilon}[\ell(\epsilon, f_c(x_t, t))] \tag{12}$$

By the one-sided Chebyshev inequality,

$$
\begin{aligned}
Pr(L_c - \mathbb{E}_c[L_c] \geq a) &\leq \frac{\sigma^2}{\sigma^2 + a^2} \\
Pr(L_c \geq \mu + a) &\leq \frac{\sigma^2}{\sigma^2 + a^2}
\end{aligned}
\tag{13}
$$

where $\sigma^2 = Var[\mathbb{E}_{t,x_0,\epsilon}[\ell(\epsilon, f_c(x_t, t))]]$. Let $a = \sigma\sqrt{\frac{1-\delta}{\delta}}$, eq. (13) gives

$$Pr\left(L_c \geq \mu + \sigma\sqrt{\frac{1-\delta}{\delta}}\right) \leq \delta \tag{14}$$

and therefore, with probability $1-\delta$, the condition-specific loss is bounded by $\mu + \sigma\sqrt{\frac{1-\delta}{\delta}}$. Although we do not control on the variance, $\sigma \leq \mu$ empirically (see appendix B.1 for details of standard error analysis) and therefore, the condition-specific loss is bounded by

$$
\begin{aligned}
\mathbb{E}_{t,x_0,\epsilon}[\ell(\epsilon, f_c(x_t, t))] &\leq \mu + \sigma\sqrt{\frac{1-\delta}{\delta}} \\
&\leq \mu + \mu\sqrt{\frac{1-\delta}{\delta}} \\
&\leq \left(1 + \sqrt{\frac{1-\delta}{\delta}}\right)\mu \\
&= \mathcal{O}(\mu)
\end{aligned}
\tag{15}
$$

therefore the per-condition loss is $\mathcal{O}(\mu)$ with high probability. $\square$

## B Additional experiments results

### B.1 Empirical results amortization loss and error analysis

To support the probabilistic generalization argument in appendices A.1 and A.2, we plot the trajectory of the training amortized loss mean with its standard deviation over mini-batches throughout

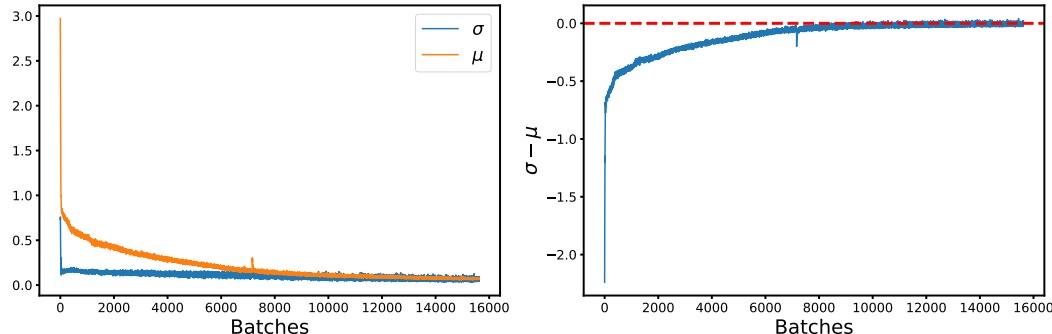

Figure 4: (left) The mean training loss of each batch is consistently above the standard deviation. (right) The difference between the standard deviation and the mean confirms that the standard deviation is consistently less than or equal to the mean.

Table 4: Ablation studies with 4 different architectural setups on the breast cancer dataset; test score is the average over all proteins.

| Method | $r$ |
|---|---|
| Best Protein Predictor | 0.489 |
| Single channel | 0.667 |
| Multi channel | 0.596 |
|    w/o output channel attention | 0.581 |
|    w/o UNet channel attention | 0.541 |
| Condition injection w/ element-wise addition | 0.516 |
| Condition injection w/ channel attention | 0.535 |
| Base (unconditional) | -0.017 |

training. The goal is to empirically validate the bounded loss assumption (Assumption 1) and the assumption that the standard deviation of the loss remains small relative to the mean.

As shown in Figure 4, the empirical standard deviation remains consistently lower than the mean across all training batches. This indicates that the loss distribution across conditioning configurations is stable and tightly concentrated, justifying our use of probability bounds on the condition-specific loss based on amortized loss. This empirical observation confirms that the generalization bound derived in appendix A.2 holds in practice, with the per-condition loss reliably bounded within a small multiple of the amortized training loss.

## B.2 ABLATION STUDIES

We conduct ablation experiments to disentangle the contributions of each architectural component. Using the breast cancer IMC dataset, we compare two feature injection mechanisms: elementwise addition and soft channel-wise attention, as well as three attention configurations: (1) with channel-wise attention only in the UNet blocks, (2) with channel-wise attention only in the output block, and (3) the full model combining both mechanisms.

Table 4 summarizes the contributions of each architectural component. The unconditional baseline performs poorly, with Pearson $r < 0.1$. Adding hierarchical feature injection significantly improves performance, and conditional injection via soft channel attention further improves over conventional element-wise addition. Output-space channel attention provides modest gains, whereas latent-space channel attention achieves more substantial improvements, capturing complex inter-channel dependencies in the latent feature space. These results support the integration of both spatially aligned conditioning and channel-wise attention for multi-channel imputation.

### B.3    ADDITIONAL RESULTS AND VISUALIZATION OF SPATIAL PROTEOMICS AND MRI GENERATION SAMPLES

We performed an empirical uncertainty analysis. For each of 500 test cases, we generated 5 independent samples and computed pixel-wise variance as a measure of predictive uncertainty. The mean variance was $0.003$, indicating highly consistent predictions. In addition, we computed local Pearson correlation with a sliding $5{\times}5$ window to quantify spatial accuracy. Together, these form a pixel-level accuracy–uncertainty map (Figure 5), where normalized variance ranges from 0 to 1 and Pearson $r$ from $-1$ to 1. The empirical distribution concentrates in the high-correlation, low-variance region (top right), showing that most pixels are both accurate and confident. Within this region, we observe the expected negative relationship: as local correlation decreases, variance systematically increases, providing a well-calibrated link between predictive accuracy and uncertainty.

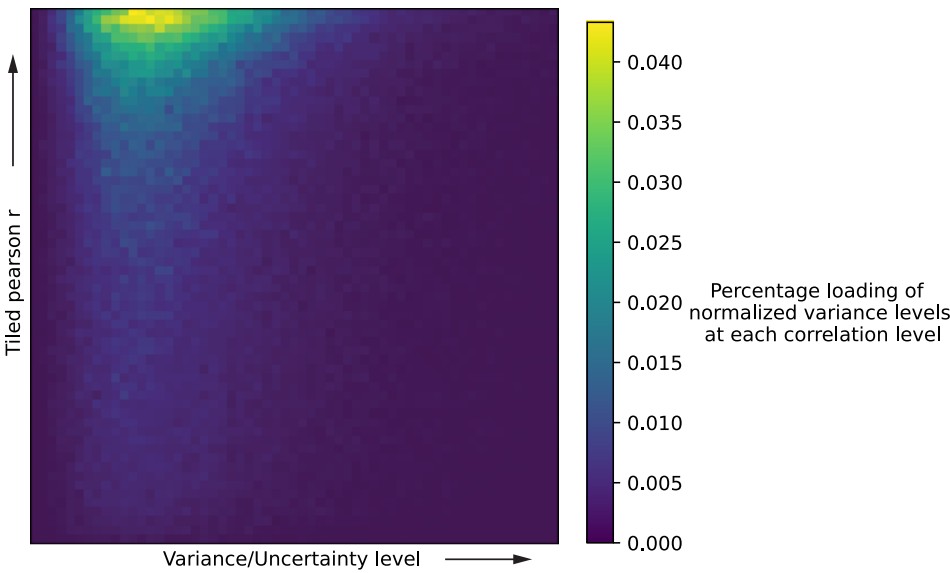

Figure 5: Visualizations of the relationship of generation accuracy and sample uncertainty.

To assess whether our model captures biologically meaningful relationships between protein channels, we evaluated the correlation between channels of the generated images. Specifically, we computed the pairwise Pearson correlation matrix across channels on (1) the generated outputs and (2) the ground truth images from the original data. We then compared these correlation matrices to assess how well the joint channel distribution is preserved. The Pearson correlations between the generated and ground-truth correlation matrices are $r = 0.925$ ($MSE = 0.031$) in the lung dataset and $r = 0.892$ ($MSE = 0.037$) in the breast dataset. We found that the correlation structure in the generated data closely mirrors that of the ground truth, demonstrating that our model not only accurately imputes each channel but also retains the inter-channel dependencies critical for downstream biological interpretation.

Figure 6 shows high-resolution generated samples of the nucleus marker (DNA1), tumor markers (PanCK, Ki67, MMP9), immune markers (CD3, CD4, CD8a, CD45RO, FOXP3, CD68), and stromal markers (CD31, CD140b), confirming the generation quality and fidelity of our model.

Figure 7 shows the generated MRI samples for all 4 MRI signals (T1CE, T1N, FLAIR, and T2W), suggesting our model generalizes well to other spatial biological data.

Table 5: Comparison of predictive performance across ControlNet and BrushNet baselines, as well as hybridized variants in which control modules are reused with our denoising backbone.

| Method | Breast | Lung |
|---|---|---|
| ControlNet (Zhang et al., 2023) | 0.452 | 0.537 |
| BrushNet (Ju et al., 2024) | 0.357 | 0.394 |
| ControlNet hybrid | 0.516 | 0.583 |
| BrushNet hybrid | 0.486 | 0.541 |
| Ours (single channel ) | **0.667** | **0.703** |
| Ours (multi channel) | 0.596 | 0.647 |

### B.4 HYBRIDIZATION EXPERIMENT OF BENCHMARKING DIFFUSION-BASED ALGORITHMS

To better compare the performance of our model with the SOTA diffusion-based method, we test both ControlNet and BrushNet as the baselines, with additional experiments with hybrid architectures, where we combine the conditional network in ControlNet (Zhang et al., 2023) and BrushNet (Ju et al., 2024) with our diffusion branch.

We evaluate each model on two IMC datasets (Breast and Lung), measuring the performance with Pearson $r$ correlation. "ControlNet hybrid" refers to a model that reuses the ControlNet conditioning branch but replaces the main denoising network with our backbone, which includes the channel attention mechanism we proposed. Similarly, "BrushNet hybrid" would replace BrushNet's denoising path with ours, in which the model has spatial attention in the UNetBlocks. Both hybrid models use element-wise addition for injection.

As shown in Table 5, ControlNet outperforms BrushNet. Notably, BrushNet's UNetBlock does not have either a channel or spatial attention module, whereas ControlNet incorporates spatial attention. The extra spatial attention may help the performance of ControlNet, even though BrushNet claims to have a better performance in natural image tasks. However, BrushNet's performance is not ideal even in the hybrid setup, which introduces spatial attention in the UNetBlocks, suggesting over-conditioning. Our full model, trained jointly using random masking, outperforms both baselines and hybrid models by a large margin.

## C MODEL ARCHITECTURE AND EXPERIMENTS SETUP

### C.1 MODEL ARCHITECTURE

We implement our model based on the EDM framework (Karras et al., 2022), modified and adapted for conditional generation under the multichannel setup. The diffusion branch has a 3-level UNet with downsampling and upsampling paths. Each UNetBlock contains a spatial attention module followed by a channel attention module. Time conditioning is implemented via sinusoidal positional embeddings.

Spatial conditions are processed by the conditional network, which has an architecture identical to the diffusion branch. Conditional features are injected into the diffusion network in middle blocks and skip connections, following ControlNet, but with soft channel attention instead of element-wise addition. We also implement the model such that it can inject the conditional features at each resolution level, like BrushNet, but as shown in table 5, the performance is suboptimal. The output layer contains a $3 \times 3$ convolutional network to map from the latent space to the protein space, followed by the output channel attention layer.

### C.2 DATA PROCESSING

The raw multichannel IMC data are processed using the Yeo-Johnson transformation to adjust the signal strength and normalized to $[0, 1]$ by the $1^{\text{st}}$ and $99^{\text{th}}$ percentiles of each channel. The processed IMC data are partitioned into $64 \times 64$ patches. Note that our model only works on data in the original data space since we need the alignment between the same channel across multiple datasets for the multi-dataset joint training. With a pre-trained MedVAE (Varma et al., 2025) finetuned on IMC

single-channel images, which encode single-channel images to lower resolution, we can compress the size of the IMC images by a factor of 4 for each channel independently. By doing so, we get $C \times 16 \times 16$ IMC data where each channel maintains its original protein semantics. The training and test splits are grouped by different patients.

The single-cell CITE-seq data are also processed with the Yeo-Johnson transformation to adjust the signal strength and normalized to $[0, 1]$ by the $1^{st}$ and $99^{th}$ percentiles of each channel.

### C.3 LOSS AND TRAINING SETUP

The model is trained based on the noise-prediction objective. For each sample $x_0$, we generate a noisy version $q(x_t|x_0)$ and learn to predict the corresponding noise $\epsilon$, given the spatial condition $c$ and timestep $t$. The training loss is:

$$\mathbb{E}_{c \sim P(c)} \mathbb{E}_{t,x_0,\epsilon} \left[ ||\epsilon - \epsilon_\theta(x_t, t, c)||^2 \right] \tag{16}$$

where $c \in \mathcal{C}$ denotes a random subset of observed channels.

For single-channel generation tasks, one target channel is masked out, and the loss is evaluated on the target channel only. For multichannel generation, the channels are sampled by independent $Bern(0.9)$, and the loss is evaluated on the generated full-channel data. The general training hyper-parameters are:

- **Optimizer:** Adam with $\beta = (0.9, 0.999)$
- **Learning rate:** $1 \times 10^{-4}$
- **Batch Size:** 256
- **Noise Schedule:** EDM noise scheduling

## D COMPUTATION RESOURCES

All experiments were trained on NVIDIA A5000 GPUs with 24G RAM. The model was trained for 2000k images with a batch size of 256, taking approximately 2 hours to complete with $16 \times 16$ resolution. All results were obtained using a single-GPU setup unless otherwise specified.

All models were implemented in PyTorch based on the EDM framework (Karras et al., 2022) and trained using standard FP32 precision without mixed-precision.

## E DISCLOSURE AND STATEMENTS

### E.1 LARGE LANGUAGE MODEL USAGE

LLMs were used to check grammar and spelling and polish wording.

### E.2 ETHICS STATEMENT

All authors of submitted papers have read the Code of Ethics and adhere to it.

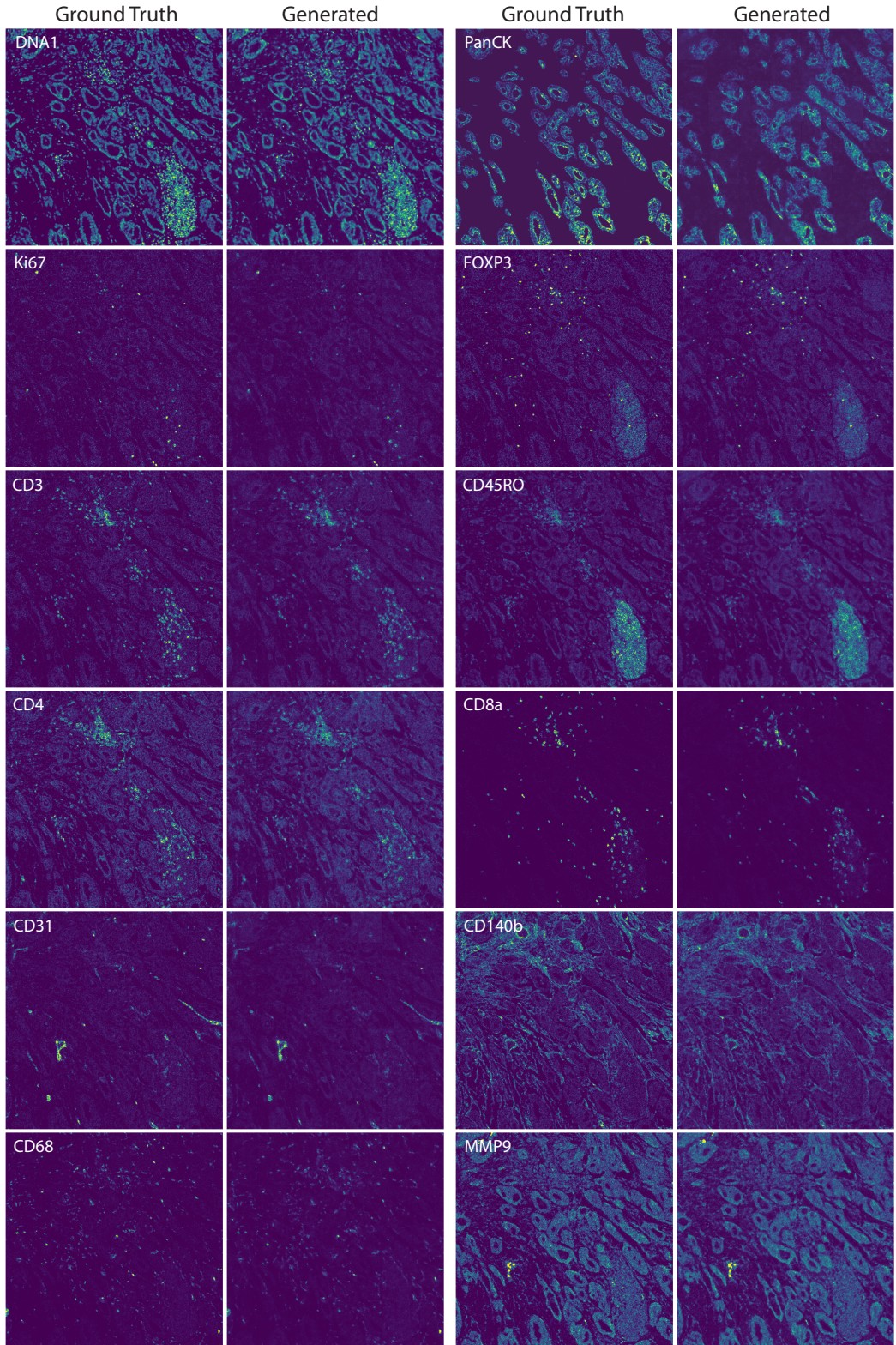

Figure 6: Visualizations of generated spatial proteomics samples of tumor and immune marker proteins of the tumor microenvironment.

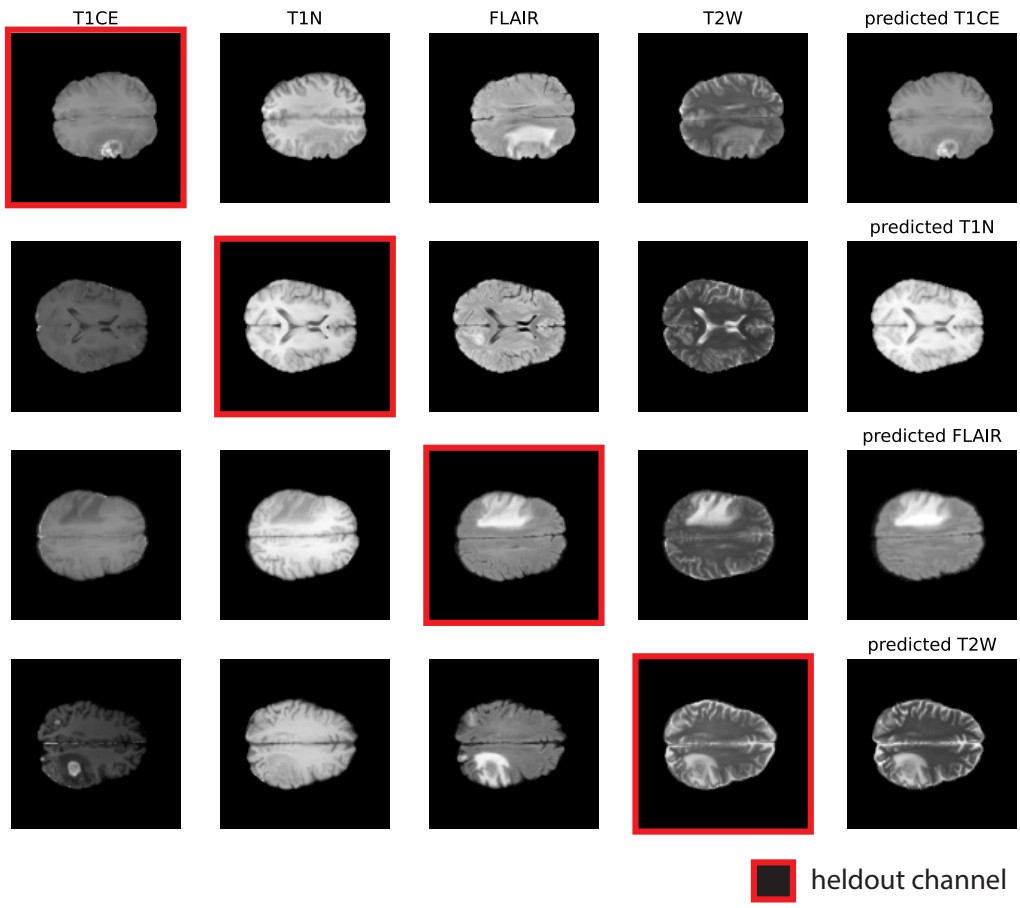

Figure 7: Visualizations of generated MRI samples for each of the MRI signals.

