# OpenReview forum: "Controllable diffusion-based generation for multi-channel biological data"
_ICLR.cc/2026/Conference — ICLR 2026 Poster_

### Official Review · Reviewer_t6Ur · 2025-10-18

**Soundness:** 2
**Presentation:** 3
**Contribution:** 2
**Rating:** 4
**Confidence:** 3

**Summary:**

Many biological modalities are inherently multi‑channel and spatially co‑registered (IMC, clinical imaging, spatial transcriptomics). Off‑the‑shelf conditional diffusion often assumes low‑dimensional channels and crude conditioning that breaks spatial correspondence. The paper proposes MCD (multi‑channel diffusion) for controllable generation and imputation given arbitrary subsets of observed channels. Core claims: (1) hierarchical feature injection to keep spatial alignment; (2) two channel‑attention modules to model inter‑channel dependencies; and (3) random channel masking for amortized training across any condition configuration. Experiments cover single‑cell gene‑to‑protein prediction, IMC imputation, and cross‑dataset generalization.

**Strengths:**

- The hierarchical feature‑injection pipeline preserves spatial correspondence at every resolution, which is exactly what biological channels need. The injection is simple to deploy and avoids brittle concatenation tricks common in vision diffusion.
- Pairing SE gating with transformer‑style channel attention gives a plausible division of labor: per‑sample recalibration and higher‑order inter‑channel structure. The additional SE head at the output enforces cross‑channel coupling where it matters.

**Weaknesses:**

- Comparators underpowered for spatial tasks. ControlNet is a fair reference, but the paper doesn’t benchmark strong end‑to‑end, jointly‑trained spatial conditioners or ablate cross‑attention conditioning versus the proposed SE‑gated injection. Given that the method already computes $E_{\ell}(c)$, a direct cross‑attention baseline would be informative.
- Masking policy and distribution shift. Random masking zeros unobserved channels in the condition (Algorithm 1). The paper does not analyze whether zero‑masking creates a train/test mismatch when “missing” at test time means “physically unmeasured but nonzero distribution,” especially for modalities where absolute intensity carries semantics.
- Metrics don’t reflect calibration. Spatial evaluation is Pearson r. No per‑channel calibration (e.g., bias/variance decomposition), no uncertainty quality, no region‑level histograms for key biomarkers. For clinical‑adjacent imaging, correlation alone can be misleading.

**Questions:**

- (Multi-channel vs multi-source) The setting is similar to multi-source integration problem, and the masking idea is similar to scVAEIT. It would be interesting to discuss the connection. (Du, J.‑H., Cai, Z., Roeder, K. (2022), Robust probabilistic modeling for single‑cell multimodal mosaic integration and imputation via scVAEIT.”)
- (Masking policy) What masking probability $p$ and schedule work best? Any performance cliffs when masks are extremely sparse/dense? Show sensitivity curves and OOD mask combos not seen in training.
- (Scalability) Provide computational complexity (e.g., wall‑clock, memory, and FLOPs) versus $C$, $D$, and $H\times W$ for both attention modules. Where does channel self‑attention become the bottleneck?

---

> ### Author Response · Authors · 2025-11-27
>
> We appreciate the reviewer’s thoughtful comments on evaluation metrics, masking policy, and scalability. We will address these concerns with both conceptual justification and additional experiments.
> 1. *Direct cross‑attention baseline*\
> The “direct cross-attention baseline” that the reviewer suggested is a standard cross-attention applied between feature maps $E_\ell(c)$ at each latent layer. Such a baseline can indeed model interactions between the feature maps. However, there are significant limitations: 1) The standard cross-attention can only operate on pairs of feature maps. To do this, one should treat each latent channel feature map independently and apply cross-attention over all pairs of feature maps, i.e. $\binom{D_\ell}{2}$ pairs per scale. This easily becomes prohibitively expensive for realistic $D$, e.g. $D=256$ leads to 32640 cross-attention operations at each layer; 2) The enormous computational cost would still only encode pairwise relationships between latent channels, not higher-order dependencies. These two reasons are exactly why we introduce our channel attention module, which treats channels as tokens and learns their joint interactions in a single attention layer, providing an efficient mechanism to capture complex channel–channel relationships without combinatorially many cross-attention blocks over pairs of latent feature maps. The computational cost prohibits this kind of baseline.
> 2. *Masking policy and distribution shift*\
> Statistically, masked zeros and true zeros are two different statistical objects, since we do not treat masked zeros as observed targets. Since the model reconstruction loss is only computed on unmasked ground truth targets, the masks do not contribute to the posterior likelihood. Therefore, the model is able to distinguish the mask from the data likelihood and has no confusion between zero-valued real signals and masked values. In practice, following the standard EDM pipeline, all data are rescaled to [-1, 1] during training. All real zero intensities are mapped -1, where the random masking makes the signals to 0. Empirically, we do not observe any channels that have uniform 0 in any patches after the rescaling.
> 3. *Metrics*\
> We appreciate the concern about metric selection. We follow the standard practice in computational biology and medical imaging, adopting correlation scores as our primary evaluation metric. In the natural image domain, metrics such as FID compare features from an Inception-v3 network trained on ImageNet. However, since the natural image features are not directly comparable to medical and biological data, no such metric currently exists for biological images, and designing a robust, biologically grounded metric remains a significant open problem.\
> On the other hand, in the MR imaging imputation task, we do provide the biologically grounded Dice coefficient [1], which is calculated based on the downstream tissue segmentation accuracy. We also provided an uncertainty analysis in Appendix B.3, which visualizes the bias-variance trade-off as a heatmap of Pearson correlations with pixel-wise variance across different samples and random seeds.
>
> [1] Li H. B., et al. The Brain Tumor Segmentation (BraTS) Challenge 2023: Brain MR image synthesis for tumor segmentation (BraSyn). 2024 https://arxiv.org/abs/2305.09011

---

> ### Author Response · Authors · 2025-11-27
> **Part 2**
>
> Questions:
> 1. *Multi-channel vs multi-source*\
> Thank you for highlighting the connection to the multi-source integration setting. This perspective naturally broadens the scope of our method. Our problem setup in the IMC imputation task is indeed related to multi-source integration: different panels and modalities provide partially overlapping views of the same underlying tissue. Our contribution is to cast this as a controllable diffusion problem, where a single generative model integrates these sources and supports conditioning on arbitrary observed subsets to impute or generate missing channels.\
> On the other hand, scVAEIT applies a mask to the VAE input over features and predicts masked entries, which is a supervised prediction rather than a conditional generation task. In fact, masking inputs or features in an autoencoder and reconstructing them is a long-standing, generic idea dating back to denoising autoencoders and masked autoencoders in vision, where randomly masked features/patches are reconstructed as a supervised objective. The masking mechanism we use is conceptually and technically distinct from the masking in scVAEIT. Our random masking operates in the conditional space instead of the input space. We explicitly frame this as a channel-wise generalization of classifier-free guidance rather than supervised feature masking. More importantly, they are mathematically different as the VAE with input mask is modeling $p(x|x_m)$ and the conditional diffusion with random mask is modeling $p(x_{t+1}|x_t, c_m)$.  The similarity is that both methods use binary masks. However, the model class, the role of the mask, and the underlying problem are fundamentally different.
>
> 2. *Masking policy*\
> To quantify sensitivity to the choice of $p$, we performed a new grid experiment sweeping $p \in [0.90, 0.80, 0.70, 0.60, 0.50, 0.40, 0.30, 0.20, 0.10, 0.05, 0.01]$ to show that the model remains stable under different masking probabilities, including much sparser observation patterns than used in the main experiments. The table below reports the mean Pearson correlation ($\pm95$% CI) between imputed and ground-truth markers for both breast and lung cohorts. Performance degrades smoothly and monotonically as $p$ decreases (i.e., as the number of observed channels becomes more sparse). In both cohorts, results show that our default $p = 0.9$ is near-optimal and the performance is robust over a broad range $p \in [0.6, 0.9]$, and only degrades significantly when the observed subset becomes extremely small $(p < 0.1)$.
> |p|Breast|Lung|
> |-|-|-|
> |0.90|0.595$\pm$0.148|0.648$\pm$0.105|
> |0.80|0.593$\pm$0.145|0.647$\pm$0.100|
> |0.70|0.571$\pm$0.147|0.644$\pm$0.099|
> |0.60|0.553$\pm$0.148|0.631$\pm$0.101|
> |0.50|0.538$\pm$0.151|0.622$\pm$0.102|
> |0.40| 0.522$\pm$0.147|0.599$\pm$0.105|
> |0.30| 0.504$\pm$0.145|0.576$\pm$0.107|
> |0.20| 0.419$\pm$0.148|0.544$\pm$0.104|
> |0.10| 0.332$\pm$0.139|0.467$\pm$0.099|
> |0.05| 0.273$\pm$0.135|0.370$\pm$0.110|
> |0.01| 0.196$\pm$0.124|0.259$\pm$0.129|
>
> 3. *Scalability*\
> We report the computational complexity of our default hyperparameter configuration in Appendix D. Because the model a large number of tunable hyperparameters besides $C$, $D$, etc., it is hard to exhaustively benchmark runtime and memory usage over the entire hyperparameter space. Instead, we provide representative measurements below, showing how computation time and GPU memory consumption scale with the latent feature dimension $D$ with the default input dimension as in the paper with batchsize of 1024, a latent channel multiplier fixed to 1, and blocks per layer fixed to 2.
> |D|time (s/batch)|gpumem(GB)|
> |-|-|-|
> |64|1.2|3.05|
> |128|1.3|5.97|
> |256|1.7|12.51|
> |512|4.3|28.47|

---

> > ### Comment · Area_Chair_ew6o · 2025-11-28
> >
> > Dear Reviewer,
> >
> > Please make sure you read the authors' response and engage with them in the discussion before the end of the discussion period on **Dec 03 '25 09:00 PM UTC**. This is a hard deadline.
> >
> > Thank you for supporting quality peer review at ICLR.
> >
> > AC

---

### Official Review · Reviewer_iF6S · 2025-10-29

**Soundness:** 3
**Presentation:** 1
**Contribution:** 3
**Rating:** 2
**Confidence:** 3

**Summary:**

The paper proposes MCD, a conditional diffusion framework for multi-channel biological data. Conditioning is handled via masked images. The proposed architecture is a dual network with hierarchical feature injection from a contextual encoder into the denoiser, plus two channel-attention mechanisms (SE for injection and transformer-style channel self-attention inside unet blocks). Experiments show state of the art level results on CITE-seq protein prediction, IMC channel imputation including a union vs intersection multi-dataset study, hybrid controls versus ControlNet/BrushNet, and BraTS MRI.

**Strengths:**

- **Problem relevance.** Training with random channel masking yields one model that accepts arbitrary observed subsets and making it flexible. The union and intersection result supports cross-dataset integration under partial channel overlap.

- **Strong empirical results.** When reported, the method consistently outperforms baselines. Experiments are broad and span single/multi dataset setups and including hybrid controls.

- **Ablations.** Stepwise ablations and ControlNet/BrushNet hybrids help attribute gains to hierarchical injection.

**Weaknesses:**

- **Subset-size stress-tests are missing.** One of the core claims is robustness to arbitrary observed subsets, but there is no sweep of performance vs. #observed channels / masking-probability p, nor targeted leave a group out per channel families. Single vs multi channel and union vs intersection is positive but partial.

- **Efficiency evidence.** Table 1 lists SiD(1-step) with near identical accuracy and claims two orders of magnitude speedup, but there are no wallclock analysis for readers to observe the real performance gains.

- **Method clarity.** I find the pieces in the methods section hard to put together. It requires stitching together several sections to reconstruct the exact flow of input, context encoder, SE-gated injections per scale, denoiser block attention, output SE flow. A single forward schematic/pseudocode can improve the clarity significantly.

- **CFG analogy is conceptual.** I could not directly link random masking with CFG. Instead, the authors can provide a deeper the analysis of how the compared baselines actually operate and where they fall short.

- **Reproducibility.** The paper promises code upon publication without anonymous repo or supplementary materials. This limits the verification significantly.

**Questions:**

See the questions and actionable items below. I find the core idea is strong, but the draft feels rushed; clearer presentation and a few added analyses would better show the paper's potential.


- **MedVAE ablation.** It would be nice to see the effect of the latent space choice on the results. One can convert each channel into grayscale (or stack each channel 3 times to imitate RGB) and run through a stable diffusion VAE pipeline do the latent diffusion.

- **Efficiency plots.** Single step generation has a clear advantage from its strong results and efficiency but it would be nice to see more details on how you distilled together with the performance comparisons with other baselines.

- **Confidence intervals.** Report mean $\pm95\%$ CI over multiple seeds for all main tables to quantify variance from training and random masking.

### Mistakes in text.

- **Section 2.2** The part describing classical RGB image imputation and classical RGB colorization problems are not correct. If we have $C_m = 0$ and $C_o = C = 3$ there is nothing to impute. Additionally, for the colorization example $C_m = 3$ and $C_o = 1$ implies $C = C_m + C_o = 4$ which is clearly not RGB.

- **Dataset reference for BMMC.** Table 1 report BMMC (bone marrow mononuclear cells) as BMNC both in the body and in the caption.

- **Different acronym on Table 3.** Throughout the paper the authors refer their method as MCD however Table 3 uses DiffuseMRI

---

> ### Author Response · Authors · 2025-11-27
>
> We appreciate the reviewer’s comments on masking policy, statistical significance, and clarity. We will address these concerns with both conceptual justification and additional experiments.
> Weakness:
> 1. *Subset-size stress-tests*\
> To quantify sensitivity to the choice of $p$, we performed a new grid experiment sweeping $p \in [0.90, 0.80, 0.70, 0.60, 0.50, 0.40, 0.30, 0.20, 0.10, 0.05, 0.01]$ to show that the model remains stable under different masking probabilities, including much sparser observation patterns than used in the main experiments. The table below reports the mean Pearson correlation ($\pm95$% CI over samples) between imputed and ground-truth markers for both breast and lung cohorts. Performance degrades smoothly and monotonically as $p$ decreases (i.e., as the number of observed channels becomes more sparse). In both cohorts, results show that our default $p = 0.9$ is near-optimal but that performance is robust over a broad range $p \in [0.6, 0.9]$, and only degrades substantially when the observed subset becomes extremely small $(p < 0.1)$.
> |p|Breast|Lung|
> |-|-|-|
> |0.90|0.595$\pm$0.148|0.648$\pm$0.105|
> |0.80|0.593$\pm$0.145|0.647$\pm$0.100|
> |0.70|0.571$\pm$0.147|0.644$\pm$0.099|
> |0.60|0.553$\pm$0.148|0.631$\pm$0.101|
> |0.50|0.538$\pm$0.151|0.622$\pm$0.102|
> |0.40| 0.522$\pm$0.147|0.599$\pm$0.105|
> |0.30| 0.504$\pm$0.145|0.576$\pm$0.107|
> |0.20| 0.419$\pm$0.148|0.544$\pm$0.104|
> |0.10| 0.332$\pm$0.139|0.467$\pm$0.099|
> |0.05| 0.273$\pm$0.135|0.370$\pm$0.110|
> |0.01| 0.196$\pm$0.124|0.259$\pm$0.129|
>
> 2. *Efficiency evidence*\
> The purpose of SiD (1-step) is to demonstrate that our framework is fully compatible with state-of-the-art distillation methods such as SiD, and that a multi-step sampler can be distilled into a single-step generator while preserving accuracy. The efficiency gain is already evident from the reduction in sampling steps (500 vs. 1), since one SiD step is equivalent to one sampling step in EDM-based diffusion models. Concretely, with 500 sampling steps the model requires 98.62 sec/batch for batches of 200 $43 \times16\times16$ patches, while the distilled 1-step model only needs 0.12 sec/batch for batches of the same size.
>
> 3. *Method clarity*\
> The forward schematic can be found in Figure 2, which illustrates the full end-to-end flow of the proposed model, with a zoom-in view of UNet blocks featuring sequential 1) input FiLM layer, 2) spatial self-attention, and 3) channel-wise attention modules architecture, as well as how SE-gated feature injection connects the conditional branch with the denoising branch.
>
> 4. *CFG analogy*\
> Classifier-free guidance (CFG) works by stochastically dropping conditioning via a single Bernoulli draw and jointly training both conditional and unconditional in a single model. Random channel masking generalizes this idea from a single Bernoulli to $C$ independent Bernoulli draws, one per channel. For each protein/gene, we independently decide whether it is observed and included in the conditioning set. Thus, instead of toggling “condition vs no-condition” as a whole, we generalize the idea across combinatorially many subsets of channels.
> Both approaches leverage random masking of conditions to train a single network that can represent many conditioning configurations (2 for CFD and $2^C$ for random masking). In our case, the granularity is per-channel rather than global. Appendix A formalizes this as amortized conditional training with control over per-condition losses, and Appendix B.2 provides detailed ablation studies.\
> A visual illustration of where compared baselines fall short can be found in Figure 1, where the baseline (ControlNet) performs well in channels with unitary inter-channel correlation but easily fails in those with more complex protein-protein interactions. This is due to the lack of channel attention modules and to a naive feature-injection mechanism. The contribution of each component is provided in the ablation study (Appendix B.2)

---

> ### Author Response · Authors · 2025-11-27
> **Part 2**
>
> Questions:
> 1. *MedVAE ablation*\
> The Stable Diffusion VAE is trained on 3-channel natural images and encodes each $3\times H\times W$ input to a $4\times H’\times W’$ latent. On the other hand, MedVAE is specifically trained on millions of medical images and enables single-channel compression, which is perfectly aligned with our need for dimension reduction in biological and medical imaging data. Due to the significant differences between natural image features and medical images features, we don’t think SD-VAE is a good choice for preprocessing in this case, and the dimensionality (1 from MedVAE versus 4 from SD-VAE) requires architectural modifications that lead to an unfair comparison.
> More importantly, MedVAE, as stated in Appendix C.2, serves as the dimension reduction method. In fact, the high-resolution imputation shown in Figure 6 is applied to the original image patches without MedVAE. Therefore, we don’t think the choice of MedVAE affects the soundness of our method.
>
> 2. *Efficiency plots*\
> As mentioned in our response to Weakness 2. Since our model is EDM-based, which is compatible with existing SOTA distillation methods like SiD, the SiD(1-step) is the result of direct application SiD [1]. And since none of the benchmark methods in Table 1 is diffusion-based, SiD is not applicable to them. Developing a standalone distillation method for each of the baselines is, therefore, not a fair comparison and out of the scope of this paper. On the other hand, in Table 2, the ControlNet can be distilled but does not improve performance (0.449 for breast and 0.533 for lung).
>
> 3. *Confidence intervals*\
> Here we report 95% confidence intervals for the our results in Tables 1–3, using the format $score \pm CI_{sample} (CI_{seed})$, where $CI_{sample}$ is the 95% CI across test samples and $CI_{seed}$ is the 95% CI across random seeds. The results show that the 95% lower bounds of our method are above or comparable to the mean performances of most of the strongest baselines. And the CIs over random seeds are less than 0.01, indicating that training is stable and that the conditional mechanism is robust to randomness from initialization and random masking. Additionally, we have provided a dedicated section on uncertainty analysis over different random seeds (Appendix B.3) in the initial submission.
>
> | |Breast|Lung|
> |-|-|-|
> |MCD|0.596$\pm$0.146(0.006)|0.647$\pm$0.104(0.004)|
> |ControlNet|0.452$\pm$0.151(0.005)|0.537$\pm$0.127(0.004)|
>
> | |DICE|$SSIM_\text{tumor}$|$SSIM_\text{health}$|$SSIM_\text{tissue}$|$SSIM_\text{global}$|
> |-|-|-|-|-|-|
> |MCD|0.738$\pm$0.284(0.004)|0.774$\pm$0.106(0.005)|0.631$\pm$0.135(0.003)|0.643$\pm$0.131(0.003)|0.928$\pm$0.103(0.002)|
> |HF-GAN|0.714$\pm$0.324(0.005)|0.761$\pm$104(0.003)|0.604$\pm$0.137(0.005)|0.615$\pm$0.132(0.004)|0.919$\pm$0.102(0.002)|
> |SwinUNETR|0.709$\pm$0.317(-)| 0.759$\pm$129(-)|0.628$\pm$0.132(-)|0.637$\pm$0.153(-)|0.916$\pm$0.121(-)|
>
> | |PBMC|CBMC|BMMC|HSPC|
> |-|-|-|-|-|
> |$r_c$|0.880$\pm$0.126(0.001)|0.962$\pm$0.105(0.001)|0.879$\pm$0.131(0.001)|0.865$\pm$0.127(0.001)|
> |$r_p$|0.673$\pm$0.167(0.002)|0.763$\pm$0.153(0.001)|0.685$\pm$0.184(0.002)|0.647$\pm$0.161(0.001)|
>
> Mistakes in text:
> 1. We respectfully disagree with the reviewer. In fact, $C_o=3, C_m=0, C=3$ is the exact definition of the classical natural image imputation problem, where the input is a corrupted image with all 3 channels ($C_o=3$) and the output is the uncorrupted image ($C=3$). There’s no channel missing in the natural image imputation task, and therefore $C_m=0$, while the imputation happens in the corrupted region of the input image. On the other hand, colorization tasks take a greyscale image ($C_o=1$) as input and predict an RGB image ($C_m=3$). Therefore, $C = C_o + C_m = 4$.
> 2. Thank you for pointing out the typos. We will revise the manuscript accordingly.
>
> [1] Zhou, M., Zheng, H., Wang, Z., Yin, M., & Huang, H. Score identity Distillation: Exponentially Fast Distillation of Pretrained Diffusion Models for One‑Step Generation. In Proceedings of the 41st International Conference on Machine Learning (ICML). 2024.

---

> > ### Comment · Reviewer_iF6S · 2025-11-27
> >
> > I would like to thank the authors for addressing my concerns. Here are my follow-up comments:
> >
> > > Subset-size stress-tests
> >
> > Your sweep shows a monotonic trend and peaks at p=0.9. However, p=0.95 for inspection is still a valid option when there is an obvious trend. Choice of p=0.9 is not justified from the reported table.
> >
> > > Efficiency evidence
> >
> > SiD version being faster is apparent, however the authors did not quantify this in the submission. Compatibility with no significant performance drop is important as long as the authors quantify the gain as well.
> >
> > > Method clarity
> >
> > I would like to thank the authors for their addition. Schematic greatly improves the clarity of the approach.
> >
> > > CFG analogy
> >
> > Yes CFG randomly drop the condition (a single Bernoulli), however it is not just a training time dropout. Its defining feature is inference time guidance (by mixing conditional and unconditional predictions with a scale). On the other hand, your method does not use guidance at sampling. It simply conditions on the masked input. Calling it a “generalization of CFG” blurs this key procedural difference.
> >
> > Moreover, a simple masking can only tell which condition you use, not how strongly to guide which is a fundamental difference compared to CFG. In fact, varying the guidance scale yields infinitely many guidance strengths, while masking yields $2^C$ subsets. These are orthogonal axes.
> >
> > > MedVAE ablation
> >
> > I believe it is very easy to refute my suggestion with an experiment.
> >
> > - For $x \in \mathbb{R}^{C\times H\times W}$ and each channel $x^{(c)}$, form a gray triplet $[x^{(c)},x^{(c)},x^{(c)}]$, encode and decode with SD-VAE. Then apply mean pooling the 3 decoded channels to get $\hat{x}^{(c)}$. Stack the $\hat{x}^{(c)}$ to recover $\hat{x}$.
> > - Report per-channel MSE for MedVAE vs SD-VAE on the same validation set.
> >
> > In case VAE provides a better reconstruction I do not believe changing two layers (input layer to accept $4\times$ and output channel to return $4\times$) would be a severe architectural modification and lead to an unfair comparison.
> >
> > > Efficiency plots
> > I understand it not being applicable to every other baseline. However, in this case, I recommend providing a detailed procedure on how the authors applied the distillation, which is still missing.
> >
> > > Confidence intervals
> >
> > I appreciate the additional tables.
> >
> >
> > > Mistakes in text
> >
> > - $C_o=3, C_m=0, C=3$ case: Your first example mixes spatial inpainting with channel imputation. If all three RGB channels are observed ($C_o=3$) and only a spatial region is corrupted, that’s an inpainting problem, not a missing channel problem.
> > - Colorization: Claiming $C = C_o + C_m = 4$ conflicts with the earlier definition of C as the channel count of the target data domain (RGB $\rightarrow$ $C=3$). Writing C=4 suggests a single 4 channel domain, which it is not.

---

> > > ### Author Response · Authors · 2025-12-03
> > > **Part 1**
> > >
> > > We thank the reviewer for the constructive feedback. And here we provide follow-up experiment results and clarifications.
> > > 1. *Subset-size stress-tests*\
> > > We thank the reviewer for pointing out that the sweep experiment shows a monotonic trend up to $p=0.9$. In our setting, as stated in the paper, realistic IMC datasets, including ours, have only 30-40 protein channels. Pushing the masking parameter to $p=0.95$ would make almost all training examples have at most one missing channel, and a substantial fraction of examples would have no missing channels at all. In that regime, the model effectively sees almost fully observed panels during training and rarely experiences the challenging imputation scenarios that motivate our method. This is exactly why we would assume no practitioner would consider such an extreme $p$.
> > > To justify this issue, we have run an additional experiment at $p=0.95$. The mean Pearson correlations at $p=0.95$ are $0.589 \pm 0.151$ for breast and $0.644 \pm 0.115$ for lung, which are marginally lower than the corresponding values at $p=0.9$, and with noticeably wider confidence intervals. This is consistent with the intuition above: overly conservative masking leads to a model that is not sufficiently trained on non-trivial missing-channel configurations, reducing both average performance and stability.
> > > As a rule of thumb, $p$ should be chosen high enough to provide sufficient channel for the model to exploit channel-channel relationships, but not so high that Bernoulli draws yield too many fully observed panels. We will update the manuscript to include the $p=0.95$ results together with the sweeping experiment results.
> > >
> > >
> > > 2. *Efficiency evidence*\
> > > We agree that these quantifications are helpful and will update the paper accordingly.
> > >
> > >
> > > 3. *CFG analogy*\
> > > We fully agree that a defining feature of CFG is its inference-time guidance, where conditional and unconditional predictions are mixed with a tunable guidance scale $\gamma$. Our method does not perform such mixing at sampling time. At inference, we always use the fully conditional model, which in CFG terminology corresponds to $\gamma=1$, i.e., the strongest possible guidance. This is a deliberate design choice. Our goal is controllable generation given a set of observed channels, so we want the model to respect those observations as strongly as possible.
> > >
> > >     We have to emphasize that 1) we are not claiming that CFG is a “generalization of CFG” in the paper, and 2) we clearly stated that our method is only inspired by the training-time masking intuition from CFG in that paper. In fact, CFG is only mentioned in the "Related Work" section, where we explicitly frame it as a source of conceptual inspiration. We found the idea of occasionally dropping the condition during training useful for exposing the model to a range of conditional configurations, as in “(we) adopt this principle in the form of random-masking guidance, where the spatial conditions are randomly masked during training.”

---

> > > > ### Author Response · Authors · 2025-12-03
> > > > **Part 2**
> > > >
> > > > 4. *MedVAE ablation*\
> > > > We appreciate the suggestion to evaluate a pretrained SDVAE as an alternative encoder–decoder. In the manuscript and in our initial response, we already explained why SD-VAE is poorly matched to IMC data: it is trained on dense natural RGB images, assumes 3-channel inputs, and uses an 8× spatial compression into a 4-channel latent space, whereas our MedVAE is designed explicitly for sparse, single-channel biological images and intensity statistics tuned to IMC. This architectural and data-distribution mismatch is precisely why domain-specific VAEs such as MedVAE have been developed.
> > > > To directly address your comment, we nevertheless implemented the exact experiment you proposed. We report per-channel MSE and Pearson correlation on the same validation sets. We considered two settings: 1) the standard SDVAE resolution ($8\times$ compression), 2) an “upscaled” setting where we first upsample the IMC images by a factor of 2 so that the effective compression ratio matches MedVAE’s $4\times$.
> > > >
> > > >     Because IMC images are very sparse, all MSE values are numerically small and less discriminative. The more biologically relevant metric is the Pearson correlation, where MedVAE substantially outperforms SDVAE in both datasets (e.g., $0.786$ vs. $0.608$ on breast and $0.818$ vs. $0.698$ on lung, even in the most favorable SDVAE configuration). In fact, our method’s imputation correlation is even comparable to that of SDVAE’s full-data reconstruction.
> > > >
> > > >     Regarding the statement that “changing two layers (input and output) would not be a severe architectural modification and lead to an unfair comparison”, we don't think 4 times more information would lead to a fair comparison, given SDVAE has 4 latent channels while MedVAE only has one. A comparison at equal latent dimensionality would therefore require either (i) naively averaging the 4 latent channels into one, which is not invertible and degrades reconstruction, or (ii) training an additional encoder–decoder to map the 4-channel SD-VAE latent to a 1-channel latent, which effectively defines a new model and breaks the premise of using an off-the-shelf SD-VAE.
> > > >     Even if we do not enforce equal latent dimension and allow SD-VAE to keep its 4-channel latent (i.e., giving it strictly more capacity), even with perfect imputation, one can at most get correlations $0.608$ for breast and $0.698$ for lung, which is worse than our single-channel MedVAE reconstructions and comparable to our current multi-channel reconstruction results.
> > > >
> > > >
> > > > ||Breast MSE| Breast Pearson $r$|Lung MSE|Lung Pearson $r$|
> > > > |-|-|-|-|-|
> > > > |SDVAE ($2\times$ upscaled)|0.005$\pm$0.004| 0.608$\pm$0.134|0.006$\pm$0.005|0.698$\pm$0.151|
> > > > |SDVAE (standard)|0.013$\pm$0.009|0.368$\pm$0.206|0.016$\pm$0.013|0.538$\pm$0.233|
> > > > |MedVAE (our finetuned)|0.006$\pm$0.004|0.786$\pm$0.125|0.006$\pm$0.006|0.818$\pm$0.145|
> > > >
> > > >
> > > > 5. *Mistakes in text*
> > > > - For the case $C_0=3,C_m=0,C=3$, we agree that this is image inpainting, which is precisely what we intend. We never claim this is a “missing channel” problem. In fact, as explicitly stated in both the paper and our initial rebuttal, this configuration corresponds to a pure spatial imputation setup with no missing channels. The purpose of this example is to show that our formulation in section 2.2 covers the special case, not to reclassify natural image inpainting as channel imputation. We will adjust the wording to make this explicit in the main text.
> > > > - For the colorization example, in section 2.2, we explicitly define C as the total number of all channels involved in the conditional generation task $C = C_o+C_m$. In the colorization setting, we have $C_o=1$ (grayscale) and $C_m=3$, so $C=4$. The target image domain is still RGB with 3 channels. The additional channel comes from the observed grayscale input, which we treat as an extra conditioning channel in our notation. We will update the wording to avoid misunderstanding.

---

### Official Review · Reviewer_aLRt · 2025-10-30

**Soundness:** 1
**Presentation:** 1
**Contribution:** 1
**Rating:** 0
**Confidence:** 4

**Summary:**

This work proposes a unified multi-channel diffusion (MCD) framework for controllable generation of structured biological data.

**Strengths:**

1. The idea of developing a framework capable of controllably generating multi-channel biological data using diffusion models is interesting.

**Weaknesses:**

1. The paper is quite obscure and its objective remains unclear. The title suggests that it focuses on developing a generative framework for multi-channel biological data, but the type of data is not specified. I assumed the authors were referring to images, yet in the experiments they attempt to predict protein expression from paired scRNA-seq data, and later they evaluate their method on MRI images. This inconsistency makes the overall methodology difficult to understand and significantly undermines the coherence of the paper.

2. The paper contains several incorrect claims and assertions. To list a few: “Existing generative models typically assume low-dimensional inputs (e.g., RGB images)” and “In spatial profiling data, each channel designates a specific molecule of interest (e.g., proteins n ≥ 30 and genes n ≥ 100), and each pixel (or cell)…”.

3. The manuscript contains numerous writing inconsistencies, redundant and buzzword-heavy claims, and incorrect or oversimplified descriptions of diffusion theory. Core method elements (random channel masking, SE attention) are incremental and poorly justified as novel contributions.

**Questions:**

1. The authors declared the following: "All experiments were trained on NVIDIA A5000 GPUs with 24 GB of VRAM. The model was trained for 2000k imgs with a batch size of 256, taking approximately 2 hours to complete at 16 × 16 resolution. All results were obtained using a single-GPU setup unless otherwise specified." Which kind of biological data have a resolution of 16 × 16?

---

> ### Author Response · Authors · 2025-11-27
>
> 1. In section 2, we explicitly state that the focus is on structured multi-channel biological and medical data, not “images” in the narrow natural-image sense. In the second paragraph of Section 2.2, we formally define this setting and explain how it covers several important special cases depending on the spatial dimensions (H, W):
>     1. when H=W=1, the model reduces to a vector-valued setting such as single-cell expression (e.g., CITE-seq RNA+protein);
>     2. when H,W>1, it covers spatial profiling data such as spatial proteomics and transcriptomics; and
>     3. the same formulation naturally extends to medical imaging modalities (e.g., MRI) with multi-channel volumes.
>
>     In other words, the number and semantics of channels are the unifying focus, while the spatial structure simply specifies which subcase of the framework we are in.
>     Our experimental design is therefore intentional rather than inconsistent. We apply the same generative framework to one representative dataset from each of these categories: CITE-seq (non-spatial structure, H=W=1), IMC for spatial proteomics (multi-channel spatial profiling), and MRI for medical imaging. This is meant to demonstrate that the method is modality-agnostic across multi-channel biological data, not restricted to natural images.
>
>     It is true that our implementation builds on a class of architectures originally developed for images (UNet-style backbones). For imaging modalities like IMC and MRI, good performance is therefore expected. However, a key strength of our approach is that the same architecture and training objective remain effective even when there is no spatial structure for the UNet to exploit, as in the CITE-seq setting. We view the strong results on CITE-seq as evidence that our formulation captures the multi-channel structure of the data, rather than relying solely on image-like inductive biases.
>
> 2. Our statement that "existing generative models typically assume low-dimensional inputs  (e.g., natural images)" reflects the dominant practice in deep generative modeling. Canonical diffusion and GAN work is overwhelmingly developed and benchmarked on 3-channel natural images, such as ImageNet, LSUN, and face datasets like FFHQ. For example, Stable Diffusion’s autoencoder is explicitly trained to encode 3-channel images into 4-channel latent features and decode them into 3-channel outputs (section 3.1 in [1]), and EDM/ADM models are trained on natural image datasets such as CIFAR-10 and ImageNet [2, 3]. Even papers critiquing evaluation metrics for generative models note that “a wide range of natural images… were all 3-channel RGB,” underscoring that the standard setting for “existing” generative models is low-channel image data [4]. Therefore, existing generative models indeed typically assume inputs with a very low number of channels, concretely 3. In contrast, the multi-channel biological data that our method works on has a number of channels ranging from 4 to >10000, which is far beyond the existing model’s capacity.
>
>     Similarly, the statement “In spatial profiling data, each channel designates a specific molecule of interest (e.g., proteins n ≥ 30 and genes n ≥ 100), and each pixel (or cell)…” reflects common practice in biological profiling. Spatial proteomics technologies routinely use protein panels on the order of 30–60 markers [5], and spatial transcriptomics platforms commonly profile hundreds to thousands of genes (e.g., ~500–5,000 genes in fluorescence-based in situ platforms such as Xenium and >10,000 genes in probe-based platforms such as Visium) [6]. In single-cell data such as CITE-seq, each observation can include thousands of gene features alongside tens to hundreds of protein channels [7], making the effective dimensionality even higher than in spatial profiling.
>
>     If the reviewer believes these factual summaries are inaccurate, we would welcome specific counterexamples or justifications. Otherwise, we do not see any factual error in these statements.
>
> [1] Rombach, R. et al. High-Resolution Image Synthesis with Latent Diffusion Models. CVPR 2022.
>
> [2] Nichol, A.Q. & Dhariwal, P. Improved Denoising Diffusion Probabilistic Models. ICML 2021.
>
> [3] Karras, T., Aittala, M., Aila, T. & Laine, S. Elucidating the Design Space of Diffusion-Based Generative Models. NeurIPS 2022.
>
> [4] Stein, G. et al. Exposing Flaws of Generative Model Evaluation Metrics and Their Unfair Treatment of Diffusion Models. NeurIPS 2023.
>
> [5] Giesen, C. et al. Highly multiplexed imaging of tumor tissues with subcellular resolution by mass cytometry. Nature Methods, 11(4), 417–422. 2014.
>
> [6] 10x Genomics. (2023). Xenium In Situ Gene Expression User Guide (CG000582, Rev B). 10x Genomics. Available at the 10x Genomics support site: Xenium In Situ Gene Expression documentation.
>
> [7] Stoeckius, M. et al. Simultaneous epitope and transcriptome measurement in single cells. Nature Methods, 14(9), 865–868. 2017.

---

> > ### Author Response · Authors · 2025-11-27
> > **Part 2**
> >
> > 3. Regarding novelty,  we explicitly acknowledge that SE-style attention is not a novel building block in isolation. It is only one component of the condition injection mechanism. We do not present it as a standalone architectural invention. The contribution lies in:
> >     1. Formulating amortized conditional training for multi-channel biological data using random channel masks, and giving a theoretical argument that this empirical risk controls both amortized and per-condition losses (Appendix A).
> >     2. Designing a hierarchical feature injection scheme with channel-wise attention suited to high-dimensional, semantically aligned channels, and empirically showing, with ablations in Appendix B.2,  that either removing channel-wise attention or replacing structured injection with naive conditioning consistently degrades imputation performance.
> >
> > Question
> > 1. As clearly stated in Appendix C.2, the IMC data is partitioned into $64 \times 64$ patches and then downsampled by MedVAE by a factor of 4 independently for each channel, yielding a $C \times 16 \times 16$ input. This aligns with common practice in both medical imaging and natural-image diffusion models for computational reasons. The underlying biological data (IMC images, MRI scans, etc.) naturally have higher spatial resolution. They are preprocessed into $16 \times 16$ feature maps independently for each channel before being fed to the diffusion model.

---

> > > ### Comment · Area_Chair_ew6o · 2025-11-28
> > >
> > > Dear Reviewer,
> > >
> > > Please make sure you read the authors' response and engage with them in the discussion before the end of the discussion period on **Dec 03 '25 09:00 PM UTC**. This is a hard deadline.
> > >
> > > Thank you for supporting quality peer review at ICLR.
> > >
> > > AC

---

### Official Review · Reviewer_WNmC · 2025-10-31

**Soundness:** 3
**Presentation:** 3
**Contribution:** 3
**Rating:** 8
**Confidence:** 3

**Summary:**

The authors introduce a Multi-Channel Diffusion (MCD) framework, designed for controllable synthesis and imputation of multi-channel biological data such as spatial proteomics, single-cell omics, and MRI. This framework integrates a mechanism for hierarchical spatial feature injection with a dual channel-attention module. This enables the resulting models to preserve spatial alignment while capturing complex inter-channel dependencies. A wide range of experiments on publicly available benchmarks suggest that MCD outperforms existing baselines.

**Strengths:**

1) This paper combines two different methodological innovations in a quite ingenious way, effectively addressing the spatial and inter-channel complexity of biological data.

2) The resulting models demonstrates versatility across multiple domains, including spatial proteomics, single-cell omics, and MRI modality synthesis, showing strong generalization and scalability.

3) Finally, the presented evaluation is quite comprehensive. I particularly appreciate the ablation studies that assess the individual contributions of model components.

**Weaknesses:**

1) All comparisons reported in Tables 1 to 3 lack any assessement of stastistical significance. This makes it difficult to gauge whether differences in performances are actually significant.
2) There is not biologically-grounded evaluation of the imputed data. For example, are known protein markers expressed in their corresponding cells?

**Questions:**

I would ask the authors to address the weaknesses highlighted above. Particularly:
1) Evaluate differences in predictive performance between their method and the others through appropriate statistical tests
2) Check whether expected cell-specific markers are expressed in the inputed data. You could also scale up this assessement to the pathway level, and check which biological pathways are enriched in the imputed data and if the enrichment results are biologically meaningful

---

> ### Author Response · Authors · 2025-12-03
>
> We thank the reviewer for their constructive comments and suggestions regarding uncertainty quantification and biological validation.
> 1. *Uncertainty quantification*/
> Here we report 95% confidence intervals for the our results in Tables 1–3, using the format $score \pm CI_{sample} (CI_{seed})$, where $CI_{sample}$ is the 95% CI across test samples and $CI_{seed}$ is the 95% CI across random seeds. The results show that the 95% lower bounds of our method are above or comparable to the mean performances of most of the strongest baselines. And the CIs over random seeds are less than 0.01, indicating that training is stable and that the conditional mechanism is robust to randomness from initialization and random masking. Additionally, we have provided a dedicated section on uncertainty analysis over different random seeds (Appendix B.3) in the initial submission.
>
> | |Breast|Lung|
> |-|-|-|
> |MCD|0.596$\pm$0.146(0.006)|0.647$\pm$0.104(0.004)|
> |ControlNet|0.452$\pm$0.151(0.005)|0.537$\pm$0.127(0.004)|
>
> | |DICE|SSIM_{tumor}|SSIM_{health}|SSIM_{tissue}|SSIM_{global}|
> |-|-|-|-|-|-|
> |MCD|0.738$\pm$0.284(0.004)|0.774$\pm$0.106(0.005)|0.631$\pm$0.135(0.003)|0.643$\pm$0.131(0.003)|0.928$\pm$0.103(0.002)|
> |HF-GAN|0.714$\pm$0.324(0.005)| 0.761$\pm$104(0.003)|0.604$\pm$0.137(0.005)|0.615$\pm$0.132(0.004)|0.919$\pm$0.102(0.002)|
> |SwinUNETR|0.709$\pm$0.317(-)| 0.759$\pm$129(-)|0.628$\pm$0.132(-)|0.637$\pm$0.153(-)|0.916$\pm$0.121(-)|
>
> | |PBMC|CBMC|BMMC|HSPC|
> |-|-|-|-|-|
> |$r_c$|0.880$\pm$0.126(0.001)|0.962$\pm$0.105(0.001)|0.879$\pm$0.131(0.001)|0.865$\pm$0.127(0.001)|
> |$r_p$|0.673$\pm$0.167(0.002)|0.763$\pm$0.153(0.001)|0.685$\pm$0.184(0.002)|0.647$0.161\pm$(0.001)|
>
> 2. *Check whether expected cell-specific markers are expressed in the inputed data*\
> We agree that explicitly checking cell-type–specific markers is important, and we have now performed such an analysis. Concretely, since the datasets we are working with do not have paired cell masks, we segment cells with Cellpose 3 on IMC data, using DNA1 and DNA2 as nuclear markers and PanCK as a membrane marker to obtain cell masks. For each protein channel, we fit a 2-component Gaussian mixture model to the per-cell intensities and get the “high/low” expression cutoff for each protein. We then apply the same classifier (cell mask and cutoff) on the imputed panels, focusing on canonical marker sets, to test whether expected cell-specific markers are expressed in the imputed data.
> For breast IMC, we evaluate epithelial markers PanCK, E-Cadherin, SMA, CD31, CD140b, T/B markers CD3, CD4, CD8a, CD20, CD7, CD45RO, CD45RA, myeloid markers CD68, CD11c, CD15, and functional markers FOXP3, Granzyme, Ki-67, and HLA-DR. For lung IMC, we use epithelial markers PanCK, SFTPC, CC10, SMA, CD31, T/NK/B markers CD3, CD4, CD8a, CD56, NCR1, myeloid markers CD14, CD11b, CD11c, CD16, CD33, CD68, CD163, CD206, MMP9, and functional markers HLA-DR, IFN, Granzyme B, Ki-67, GITR, TOX. The F1 scores of this classifier when applied to imputed panels are 0.83 for breast and 0.84 for lung, indicating that canonical markers remain correctly localized to their expected cell types after imputation. We will add this analysis and the pipeline description to the appendix to make the biological validation more explicit.
>
> ||Breast|Lung|
> |-|-|-|
> |PanCK|0.94|0.88|
> |SMA|0.89|0.87|
> |CD31|0.83|0.73|
> |CD140b|0.84|-|
> |E-Cadherin|0.86|-|
> |SFTPC|-|0.76|
> |CC10|-|0.84|
> |CD3|0.77|0.83|
> |CD4|0.87|0.81|
> |CD8a|0.86| 0.82|
> |CD20|0.80|-|
> |CD7|0.83|-|
> |CD45RO|0.86|-|
> |CD45RA|0.74|-|
> |CD56|-|0.77|
> |NCR1|-|0.86|
> |CD11b|-|0.90|
> |CD11c|0.84|0.89|
> |CD14|-|0.88|
> |CD15|0.80|-|
> |CD16|-|0.80|
> |CD33|-|0.96|
> |CD68|0.76|0.83|
> |CD163|-|0.84|
> |CD206|-|0.83|
> |MMP9|-|0.85|
> |HLA-DR|0.88|0.87|
> |Granzyme|0.81|0.87|
> |Ki-67|0.76|0.81|
> |FOXP3|0.85|-|
> |IFN|-|0.83|
> |GITR|-|0.80|
> |TOX|-|0.75|

---

### Meta-Review · Area_Chair_SXDW · 2026-01-03

**Summary:**

This work proposes a unified multi-channel diffusion (MCD) framework for controllable generation of structured biological data.
The paper received extremely divergent scores, ranging from 0 to 8.
The 0-point reviewer reported that the manuscript was obscure and almost incomprehensible; however, after receiving the authors’ rebuttal, this reviewer offered no further response.
The 8-point reviewer explicitly stated that the work exhibits strong novelty and that the proposed method possesses good generalizability and extensibility.

**Reviewer Concerns:**

The authors supplemented the necessary experiments to address the reviewers’ concerns and provided detailed explanations and clarifications for the doubts raised. However, the reviewers did not respond constructively.

**Reviewer Scores:**

This is a highly controversial paper: the assessments from different reviewers differ dramatically. The authors have furnished detailed rebuttals to every raised issue, and I believe several of these concerns have been satisfactorily resolved; consequently, I expect at least some reviewers will raise their scores in light of the response.

---

### Decision · Program_Chairs · 2026-01-26

Accept (Poster)